# PURIFYING TASK VECTORS IN KNOWLEDGE-AWARE SUBSPACE FOR MODEL MERGING

## ABSTRACT

Model merging aims to integrate task-specific abilities from individually fine-tuned models into a single model without extra training. In recent model merging methods, task vector has become a fundamental building block, as it can encapsulate the residual information from finetuning. However, the merged model often suffers from notable performance degradation due to the conflicts caused by task-irrelevant redundancy in task vectors. Existing efforts in overcoming redundancy by randomly dropping elements in the parameter space involves randomness and lacks knowledge awareness. To address these challenges, in this study, we propose **P**urifying TA**s**k **VE**ctors (PAVE) in knowledge-aware subspace. Concretely, we sample some training examples from each task, and feed them into their corresponding fine-tuned models to acquire the covariance matrices before linear layers. We then perform a context-oriented singular value decomposition, which accentuates the weight components most relevant to the target knowledge. As a result, we can split fine-tuned model weights into task-relevant and redundant components in the knowledge-aware subspace, and purify the task vector by pruning the redundant components. To induce fair pruning efforts across models, we further introduce a spectral rank allocation strategy by optimizing a normalized activated pruning error. The task vector purification by our method as a plug-and-play scheme is applicable across various task vector-based merging methods to improve their performance. In experiments, we demonstrate the effectiveness of PAVE across a diverse set of merging methods, tasks, and model architectures. Remarkably, when integrated on the state-of-the-art merging method EMR-Merging with RoBERTa, our method yields a significant performance improvement of 4.1% (from 80.18% w/o PAVE to 84.28%) on the GLUE benchmark, approaching to the average performance of 8 individual models, 85.55%.

## 1 INTRODUCTION

Transformer-based models, including large language models (LLMs) (Devlin et al., 2019; Liu et al., 2019; He et al., 2020; Radford et al., 2019; Touvron et al., 2023; Achiam et al., 2023) and vision models (Dosovitskiy et al., 2020; Radford et al., 2021) have found widespread applications across various domains. Supervised fine-tuning (Radford et al., 2018; Hu et al., 2022) is a common technique to enhance downstream task performance. Despite the effectiveness of multi-task learning (MTL) in acquiring versatile abilities across multiple domains (Caruana, 1997; Zhang & Yang, 2018; Vandenhende et al., 2021), its applicability will be limited when only individually fine-tuned models are accessible due to data privacy concerns or computational constraints. To address this challenge, model merging has been proposed, aiming to construct a single model by combining parameters from domain-specific models, without relying on any training or fine-tuning efforts.

Early model merging methods focus on performing weighted (Matena & Raffel, 2022) or regularized (Jin et al., 2022) averaging in the parameter space of fine-tuned models, yet the conflicts inherent in naive parameter averaging often lead to performance degradation of merged models. Subsequently, Task Arithmetic (Ilharco et al., 2022) has achieved great success by introducing task vector, defined as the difference between the fine-tuned model weights for a certain task and the base model. Since task vector can capture the update direction of each task, it allows for efficient composition of knowledge from multiple fine-tuned models while preserving their distinctive features, and thus has become a fundamental building block in the follow-up studies (Ilharco et al., 2022; Cheng et al.,

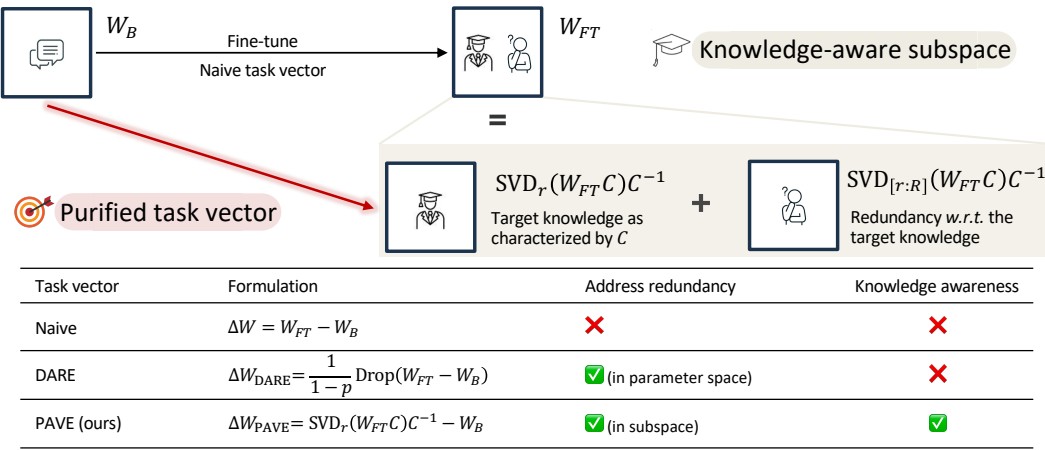

Figure 1: An illustration of our method and a comparison between PAVE and prior task vectors. DARE addresses redundancy by randomly dropping elements in the parameter space without knowledge awareness. PAVE refines task vectors by identifying the target knowledge-relevant components within knowledge-aware subspace and removing the redundant components.

2025; Du et al., 2024; Huang et al., 2024). However, the task vector constructed by subtracting the base model weights from the fine-tuned weights contains substantial redundancy, which means that not all parameter shifts during fine-tuning contribute to task-specific abilities (Yu et al., 2024). The redundant or noisy components may interfere with the valuable components from other task vectors during merging. As a result, performing model merging based on such task vectors still encounters conflicts and yields suboptimal performance (Yadav et al., 2023).

To address the redundancy generally in task vectors, plug-in methods has been proposed. DARE (Yu et al., 2024) proposes to randomly mask task vectors by Dropout (Srivastava et al., 2014) to induce sparsity and reduce conflict. Subspace Boosting (Skorobogat et al., 2025) employs High Order Generalized Singular Value Decomposition (HO-GSVD) to evaluate task similarity, and utilizes a shared subspace to distinguish between common and unique components across tasks. Despite the improved performance, we point out that: (1) random masks lack knowledge awareness and may fail to accurately locate the parameter elements most relevant to a specific task; (2) task-relevant and redundant elements are not necessarily separable in the parameter space, and actually task-specific ability lies in a low-rank subspace of parameters (Hu et al., 2022), which means that masking in the parameter space may not effectively disentangle abilities and resolve conflicts. (3) Even though HO-GSVD utilizes a shared subspace to distinguish between common and unique components across tasks, it lacks formal guarantees that the identified unique subspaces are task-aware and beneficial for merging Therefore, how to purify task vectors to make them precisely align with task-specific abilities is still a fundamental challenge in the model merging problem.

To this end, in this paper, we propose **p**urifying t**a**sk **ve**ctors in knowledge-aware subspace (PAVE). Instead of addressing redundant and noisy elements in the original parameter space, we aim to decompose fine-tuned LLM parameters into subspace such that the decomposed components are ordered by their contribution to a specific task. Naively performing singular value decomposition (SVD) on the fine-tuned weights is inadequate because it only captures the spectral structure of the weight matrix itself and the resulting subspace is not knowledge-aware. In contrast, we adopt the context-oriented decomposition that is initially proposed in a recent study for low-rank adaptation (Yang et al., 2024b). Specifically, given multiple fine-tuned models on different tasks, we randomly sample a small number of training samples for each task, and feed them into the corresponding fine-tuned model. After obtaining the activation of each layer $X$, we calculate the covariance matrix $C = XX^T$ and then perform the context-oriented decomposition by applying SVD to the product of the fine-tuned weight matrix $W_{\text{FT}}$ and the covariance matrix $C$ from its input, *i.e.,* $\text{SVD}(W_{\text{FT}}C)$. By doing so, the decomposed subspace is spanned by singular vectors that are ordered by their contributions to the task associated with $C$. It has been observed that the first several components account for the majority of the learned task-relevant knowledge. Therefore, we filter out the redundant and noisy directions by retaining only the largest $r$ singular values and their singular vectors. We then

post-multiply by $C^{-1}$ to reconstruct the fine-tuned model weights, which yields our purified task vector formulated as $\Delta W_{\text{PAVE}} = \text{SVD}_r(W_{\text{FT}}C)C^{-1} - W_{\text{B}}$. The illustration of our method and its comparison with existing task vector constructions are shown in Figure 1.

To induce fair pruning efforts across models, we further introduce a spectral rank allocation strategy that allocates the number of preserved ranks based on their spectral distribution. It aims to minimize the summation of a normalized activated pruning error from each individual model. This objective can be efficiently optimized using a greedy algorithm that iteratively excludes components with the smallest normalized singular value. It is noteworthy that PAVE as a plug-and-play method can be integrated on any task vector-based merging methods to improve their performance. We evaluate the performance of PAVE in combination with several popular merging strategies, including Task Arithmetic (Ilharco et al., 2022), Ties-Merging (Yadav et al., 2023), and EMR-Merging (Huang et al., 2024). Experiments are conducted on the GLUE benchmark with two widely used language models, RoBERTa (Liu et al., 2019) and DeBERTa (He et al., 2020), and also on generation tasks, Math and Coding, with LLaMA-2-7B (Touvron et al., 2023). Additionally, we apply our approach to ViT-B/16, ViT-B/32, and ViT-L/14 (Dosovitskiy et al., 2020) on 8 vision tasks.

The contributions of this study are summarized as follows:

- We propose PAVE, a plug-and-play method that can improve the performance of modern model merging methods by purifying task vectors and resolving task-irrelevant redundancies in knowledge-aware subspace.

- We design a spectral rank allocation strategy to further refine the redundancy pruning process and improve the performance.

- We demonstrate the effectiveness of PAVE with a wide range of merging methods, tasks, and model architectures. Notably, PAVE improves the performance of EMR-Merging on GLUE with RoBERTa from 80.18% to 84.28%, which is close to the average result of 8 individual models 85.55%.

## 2 RELATED WORK

**Model Merging.** Model merging has emerged as an active area due to its great potential in integrating abilities from multiple specialized models into a single model (Wortsman et al., 2022; Yadav et al., 2023; Yang et al., 2023; Xiao et al., 2023). Different approaches have been developed including optimization-driven methods (Matena & Raffel, 2022; Jin et al., 2022; Wang et al., 2024a), test-time adaptation methods (Yang et al., 2023; 2024a; Tang et al., 2023), and task vector-based methods (Ilharco et al., 2022; Yadav et al., 2023; Wei et al., 2025; Xiong et al., 2024). In particular, task vector has become an influential tool, as it encodes the essential information obtained during fine-tuning and can be effectively injected to a merged model using appropriate techniques. However, during the merging process, evidences have shown that there is noise and redundancy in the plain task vectors, which can cause conflicts and performance degradation. To address this issue, Ties-Merging (Yadav et al., 2023) identifies redundancy in model parameters and proposes a trimming and sign-selection scheme to resolve conflicts. EMR-Merging (Huang et al., 2024) employs a strategy of electing, masking, and rescaling to integrate lightweight task-specific components, aiming to minimize the divergence between the merged model and each individual model. TATR (Sun et al., 2025) reduces conflicts by constraining task vector combination within a trust region. DARE (Yu et al., 2024), as a plug-and-play method, resolves conflict issues by introducing random Dropout (Srivastava et al., 2014) to the task vectors. Nevertheless, former methods mainly focus on reducing redundancy by removing elements from the task vectors, which can be considered as sparsity-based approaches. In contrast, our method purifies task vectors in a knowledge-aware subspace by performing context-oriented decomposition and pruning the task-irrelevant weight components.

**Low-Rank Decomposition for LLMs.** Low-rank decomposition based on SVD has been widely adopted in LLMs across multiple applications. For example, in low-rank adaptation (Hu et al., 2022; Liu et al., 2024; Zhang et al., 2023; Yang et al., 2024b; 2025), SVD is often leveraged to create lightweight low-rank adapters for improving fine-tuning performance (Meng et al., 2024) or preserving task-specific abilities (Yang et al., 2024b; Liang et al., 2025). In model compression, various methods have been proposed to prune model parameters based on the importance provided

by singular values (Yuan et al., 2023; Wang et al., 2024b; Hsu et al., 2022; Li et al., 2025). Similar techniques have also been developed for quantization (Li et al., 2024b;a). However, when it comes to model merging, redundancy purification still relies mostly on sparsity-based approaches—removing or rescaling task vectors in the parameter space. While decomposition approaches are adopted in recent merging methods like Twin Merging (Lu et al., 2024), TSV-Merging Gargiulo et al. (2025), Doge-Merging (Wei et al., 2025), and Iso-CTS (Marczak et al., 2025), the implementations apply standard SVD to task vectors for pruning and are therefore task-agnostic. As a comparison, our proposed method performs task vector purification in knowledge-aware subspace to extract and highlight the parts within a task vector that truly matter for its corresponding task, leading to smoother and more effective merging.

## 3 METHODOLOGY

### 3.1 PRELIMINARIES

Given $K$ different models $\{\mathcal{M}\{W_1^l\}_{l=1}^L, \mathcal{M}\{W_2^l\}_{l=1}^L, \cdots, \mathcal{M}\{W_K^l\}_{l=1}^L\}$ that are fine-tuned on $K$ different tasks $\{T_1, T_2, \cdots, T_K\}$ from the same base model $\mathcal{M}\{W_B^l\}_{l=1}^L$, where $L$ represents the number of layers, model merging aims to produce a single model that simultaneously achieves good performance on these tasks without extra training or fine-tuning. The merged model $\mathcal{M}\{W_M^l\}_{l=1}^L$ can be given by:

$$\mathcal{M}\{W_M^l\}_{l=1}^L = \phi(\mathcal{M}\{W_1^l\}_{l=1}^L, \cdots, \mathcal{M}\{W_K^l\}_{l=1}^L, \mathcal{M}\{W_B^l\}_{l=1}^L), \tag{1}$$

where $\phi$ refers to a merging method.

Most model merging methods focus on combining the parameters of fine-tuned models. Task vector captures the task-specific changes introduced during fine-tuning and has become a common and essential tool in modern model merging methods. It is typically defined as the element-wise difference between a fine-tuned model and the base model,

$$\Delta W = W_{FT} - W_B, \tag{2}$$

where $W_{FT}$ denotes the fine-tuned model weights. Based on task vector, different merging methods have been proposed to integrate these task-specific parameter variations into a merged model. For example, in Task Arithmetic (Ilharco et al., 2022), the merged model weights can be computed as:

$$W_M = W_B + \lambda \sum_{i=1}^K \Delta W_i, \tag{3}$$

where $\lambda$ is a hyperparameter.

However, recent studies have shown that naively calculating the difference between $W_{FT}$ and $W_B$ cannot precisely represent the shifting direction of a certain task in the parameter space, as the weight change caused by fine-tuning contains redundant components, *e.g.,* task-irrelevant abilities or noisy elements (Yadav et al., 2023; Yu et al., 2024). As a result, there still exist conflicts in the subsequent merging procedure using these task vectors. DARE (Yu et al., 2024) proposes to sparsify task vectors by randomly dropping elements and rescaling. Their task vector can be formulated as:

$$\Delta W_{DARE} = \frac{1}{1-p} \text{Drop}(W_{FT} - W_B), \tag{4}$$

where Drop denotes the Dropout operation with rate $p$, and $\frac{1}{1-p}$ is a rescaling coefficient. Despite its effectiveness in reducing conflict, several limitations remain as follows. **First**, the Dropout operation involves randomness and lacks knowledge awareness, and thus it cannot locate the valuable components in a targeted manner. **Second**, task-specific abilities actually lie in the subspace of pre-trained parameters (Raghu et al., 2017; Li et al., 2018), suggesting that binary masking in the parameter space may not truly separate redundancy or irrelevant components from the required ability. This analysis motivates us to resort to subspace to purify task vectors.

### 3.2 PURIFYING TASK VECTORS IN KNOWLEDGE-AWARE SUBSPACE

Directly performing singular value decomposition (SVD) on fine-tuned model weights constructs a subspace, but such subspace is not necessarily aligned with task-specific knowledge, as it only

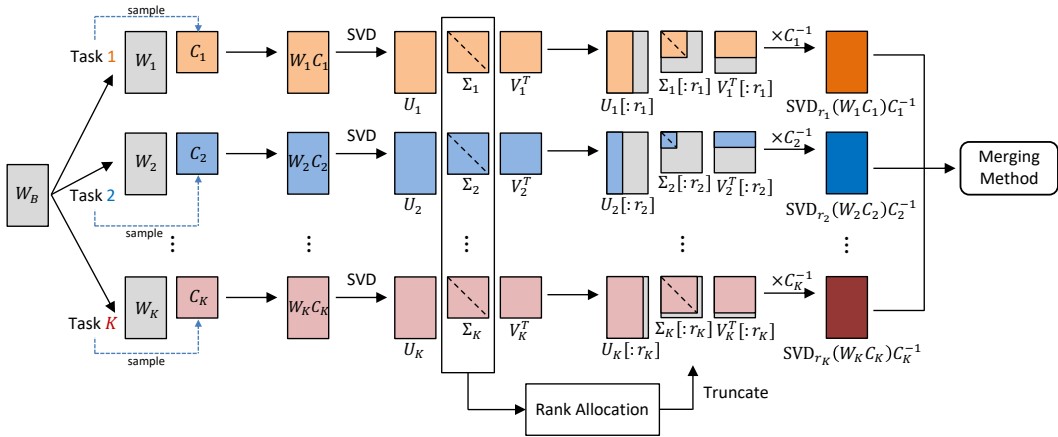

Figure 2: An overall illustration of our proposed method, PAVE, which serves as a plug-and-play method for task-vector based merging methods. It begins by identifying task-aware components through a decomposition approach, and then eliminates noise and less effective components via rank-based pruning. A rank allocation strategy is designed to optimize the preserved rank ratios across individual models. Overall, rather than employing the plain task vector $W_k - W_B$, PAVE performs model merging based on purified task vectors $\text{SVD}_{r_k}(W_k C_k)C_k^{-1} - W_B$, $1 \le k \le K$.

reflects the spectral distribution of the parameter matrix itself and pre-trained LLM weights handle a broad range of tasks. To this end, we leverage the context-oriented singular value decomposition (CO-SVD) (Yang et al., 2024b) to decompose fine-tuned weights into a knowledge-aware subspace. Concretely, for each task $T_k, 1 \le k \le K$, we sample some training examples from this task and feed them into the fine-tuned model $\mathcal{M}_k$. After acquiring the input activation of each layer $X$ (layer index is omitted for simplicity), we calculate the covariance matrix as $C = XX^T$. Projecting the weight matrix of a linear layer onto the covariance matrix of its input activation can accentuate the task-specific components related to the input. Therefore, we apply SVD on the product of each weight matrix and its corresponding covariance matrix, *i.e.,* $\text{SVD}(W_{\text{FT}}C)$. The decomposed components are ordered by the singular values that reflect their contributions to the task as characterized by $C$. Such subspace enjoys a low-rank structure, which means that the knowledge-aware abilities can be condensed into the first several leading singular values and singular vectors, as will be shown in Section 4.1. Hence, we filter out the task-irrelevant components by keeping only the first $r$ components, and then reconstruct the weight matrix through right multiplication with $C^{-1}$, *i.e.,* the inverse of covariance matrix $C$. Consequently, the purified weight matrix can be formulated as:

$$W_{\text{FT}}^{\dagger} = \text{SVD}_r(W_{\text{FT}}C)C^{-1} = \sum_{i=1}^{r} \sigma_i u_i v_i^T C^{-1}, \tag{5}$$

where $\sigma_i$ denotes the $i$-th singular value, $u_i$ and $v_i$ are the left and right singular vectors, respectively. We use the purified fine-tuned weight matrix $W_{\text{FT}}^{\dagger}$ to construct our task vector as:

$$\Delta W_{\text{PAVE}} = W_{\text{FT}}^{\dagger} - W_{\text{B}} = \text{SVD}_r(W_{\text{FT}}C)C^{-1} - W_{\text{B}}, \tag{6}$$

where $W_{\text{B}}$ is the base model weights before fine-tuning, and $\Delta W_{\text{PAVE}}$ represents our purified task vector, which can be used for any task vector-based merging methods, such as Ties-Merging (Yadav et al., 2023), Task Arithmetic (Ilharco et al., 2022), and EMR-Merging (Huang et al., 2024).

The effectiveness of CO-SVD in identifying task-specific directions of weight matrices has been verified by Yang et al. (2024b) for model adaptation. In addition, $C = XX^T$ is a symmetric positive semi-definite matrix, which enables efficient inverse operation with numerical stability. When $C$ is not invertible, we repeatedly add positive elements on its diagonal until it is invertible. In experiments, we show that our method can be integrated on various model merging methods to improve the performance of the merged models.

## 3.3 SPECTRAL RANK ALLOCATION

---

**Algorithm 1** Spectral rank allocation strategy

---

**Require:** Individual models $\{\mathcal{M}\{W_1^l\}_{l=1}^L, \mathcal{M}\{W_2^l\}_{l=1}^L, \cdots, \mathcal{M}\{W_K^l\}_{l=1}^L\}$, data sampling from target tasks $\{X_1, X_2, \cdots, X_K\}$, a preserved rank ratio $\rho$, a stopping ratio $\gamma$.

1: Feed $X_i$ into $\mathcal{M}\{W_i^l\}_{l=1}^L$. Calculate covariance matrices for all layers and for all models, $\{C_i^l\}_{i,l}^{K,L}$, and the rank of each layer $\{R^l\}_{l=1}^L$.

2: **for** $l = 1$ to $L$ **do**

3:     Apply SVD to $W_i^l C_i^l$ and compute $\{\sigma_{i,j}^l\}_{i,j}^{K,R^l}$, sort them in descending order.

4:     Normalize $\{\sigma_{i,j}^l\}_{i,j}^{K,R^l}$ by the largest singular value, $\{s_{i,j}^l\}_{i,j}^{K,R^l} = \left\{ \frac{\sigma_{i,j}^l}{\sigma_{i,\max}^l} \right\}_{i,j}^{K,R^l}$

5:     Initialize $\{r_i^l\}_{i,l}^{K,L} = \{R^l\}_{i,l}^{K,L}$

6:     Initialize index set $I = \{1, 2, \cdots, K\}$

7:     **while** $\frac{\sum_{i=1}^K r_i^l}{KR^l} > \rho$ **do**

8:         Pick up the smallest $s_{i,r_i^l}^l$ across $i \in I$, suppose it is $s_{t,r_i^l}^l$

9:         $r_t^l = r_t^l - 1$

10:        **if** $r_t^l \le \gamma R^l$ **then**

11:            remove $t$ from $I$

12:        **end if**

13:    **end while**

14: **end for**

15: Output: the rank ratios for each individual model, $\{\rho_i\}_i^K = \left\{ \frac{\sum_{l=1}^L r_i^l}{\sum_{l=1}^L R^l} \right\}_i^K$.

---

We have introduced the construction of our purified task vector in Section 3.2. However, when merging multiple fine-tuned models, assigning the rank $r$ to each model is highly non-trivial. This is because the fine-tuned individual models vary in their adaptation to rank pruning; some fine-tuned weights are more robust and can preserve performance even after aggressive rank pruning, while others are more sensitive. As a result, applying a naive rank strategy can lead to performance degradation. To optimize rank allocation under a specified preserved rank ratio $\rho$, we propose a layer-wise optimization framework that determines the preserved rank of linear layer $l$ by minimizing a summation of normalized activated pruning error from different models. This strategy enables an adaptive preservation of task-specific knowledge during the merging process.

We begin by introduce the activated pruning error with rank $r_i^l$ as:

$$\|(\Delta W_{\text{PAVE},i}^l - \Delta W_i^l))C_i^l\|_F^2 = \|\text{SVD}_{r_i^l}(W_i^l C_i^l) - W_i^l C_i^l\|_F^2, \tag{7}$$

where $i$ refers to the $i$-th model. This error estimates the activated distance between our purified task vector and the original plain task vector. Since the plain task vector retains the full capabilities through fine-tuning, this error serves as a measure of how well the purified task vector preserves task-specific knowledge. In practice, we further normalize it with $\sigma_{\max}(W_i^l C_i^l)$ to ensure comparability across different models with various spectral magnitudes. This normalization bounds the scaled spectrum within a range from the inverse of its condition number to 1, thus enabling a fair assignment of rank across models. The normalized activated pruning error is defined as:

$$\frac{\|\text{SVD}_{r_i^l}(W_i^l C_i^l) - W_i^l C_i^l\|_F^2}{(\sigma_{\max}(W_i^l C_i^l))^2} = \sum_{j=r_i^l+1}^{R^l} \frac{(\sigma_j(W_i^l C_i^l))^2}{(\sigma_{\max}(W_i^l C_i^l))^2} = \sum_{j=r_i^l+1}^{R^l} \left( \frac{\sigma_{i,j}^l}{\sigma_{i,\max}^l} \right)^2. \tag{8}$$

Building on this formulation, we define the overall objective for the $l$-th layer as:

$$\min_{\frac{\sum_{i=1}^K r_i^l}{KR^l} < \rho} \sum_{i=1}^K \frac{\|\text{SVD}_{r_i^l}(W_i^l C_i^l) - W_i^l C_i^l\|_F^2}{(\sigma_{\max}(W_i^l C_i^l))^2}, \tag{9}$$

where $R^l$ refers to full rank. Minimizing the objective Eq. (9) can be effectively approached using a greedy algorithm, and the whole procedure has been summarized in Algorithm 1. In this procedure, for each layer $l$, we iteratively remove the component associated with the smallest normalized singular value, defined as $s_{i,j}^l = \frac{\sigma_{i,j}^l}{\sigma_{i,\max}^l}$, thereby achieving optimality of Eq. (9) through the greedy

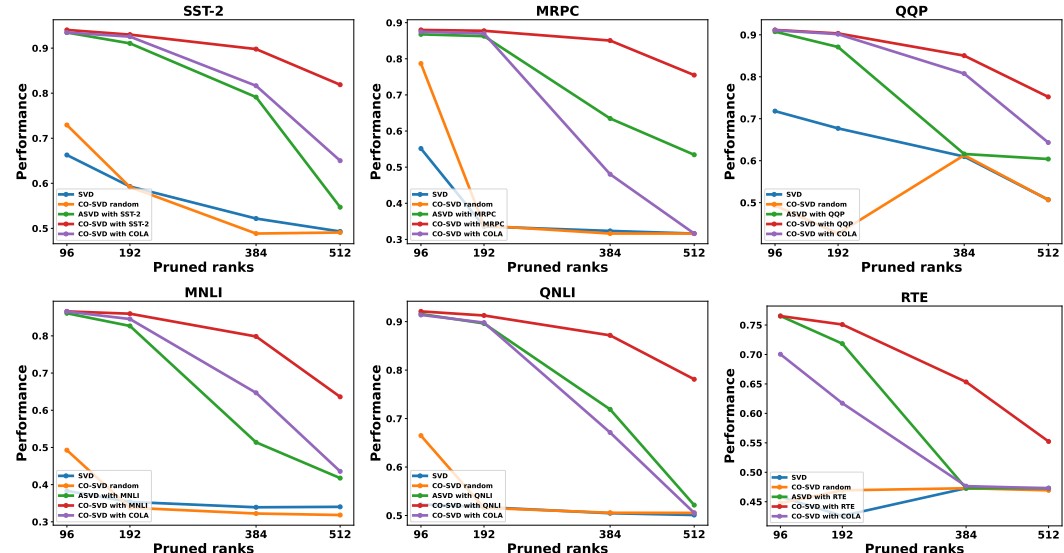

Figure 3: Model rank pruning with different decomposition methods. The results show that the context-oriented decomposition (CO-SVD) is more effective in aligning the subspace with the target knowledge. The pruned ranks, 96, 192, 384 and 512, correspond to removing 1/8, 1/4, 1/2 and 2/3 of the full rank, respectively.

process. To prevent excessive rank pruning that causes performance degradation, we introduce a threshold $\gamma$ which serves as a stopping ratio. This rank strategy enables spectrum-based rank allocation for model merging while respecting the global rank constraint.

## 4 EXPERIMENTS

In this section, we provide the results on the GLUE benchmarks using RoBERTa-base (Liu et al., 2019) and DeBERTa-base (He et al., 2020), as well as the generation benchmarks with LLaMA2-7B-hf (Touvron et al., 2023) on mathematical solving and code generation tasks. We evaluate PAVE with various merging methods, including Task Arithmetic (Ilharco et al., 2022), Ties-Merging (Yadav et al., 2023), and EMR-Merging (Huang et al., 2024). For further detailed experimental settings and hyperparameters, please refer to Appendix A.

### 4.1 EFFECT OF DECOMPOSITION IN KNOWLEDGE-AWARE SUBSPACES

We demonstrate the effectiveness of decomposition in knowledge-aware subspaces through a series of numerical experiments. As concluded in Figure 3, we prune individual RoBERTa models fine-tuned on six GLUE tasks using various decomposition methods. The decomposition methods mentioned in Figure 3 are summarized as follows: SVD (Meng et al., 2024) applies singular value decomposition directly to the weights. ASVD (Yuan et al., 2023) introduces a diagonal scaling matrix, where each diagonal entry corresponds to the average absolute value of the associated feature. CO-SVD (Yang et al., 2024b) leverages covariance matrices. In the CO-SVD random variant, the covariance matrix is replaced by a sampled random square matrix from [-1,1], serving as a task-agnostic baseline. CO-SVD with CoLA derives the covariance matrix using samples from CoLA, highlighting the impact of selecting samples from an appropriate task.

From Figure 3, it is evident that, for a given rank, decomposition methods that incorporate task information (CO-SVD, ASVD) outperform those that do not (SVD, CO-SVD random). In particular, CO-SVD demonstrates superior performance compared to ASVD, as CO-SVD preserves the context information more effectively through task-specific activations. This pattern holds consistently across all datasets evaluated, reinforcing the importance of using the correct task data to guide decomposition. Notably, performance advantages of CO-SVD become even more prominent at pruning larger ranks, where other methods exhibit sharp degradation, indicating CO-SVD's effectiveness in condensing knowledge awareness. It is noteworthy that the input must originate from

Table 1: Results of merging fine-tuned RoBERTa models on eight datasets from GLUE benchmark. The number of examples used to generate context matrix is 4096.

| Methods | CoLA | SST-2 | MRPC | STSB | QQP | MNLI | QNLI | RTE | Average |
|---|---|---|---|---|---|---|---|---|---|
| Individual | 60.18 | 94.04 | 89.22 | 90.63 | 91.41 | 87.20 | 92.71 | 79.06 | 85.55 |
| EMR-Merging | 39.96 | 93.35 | 86.27 | 82.77 | 89.72 | 85.45 | 89.57 | 74.37 | 80.18 |
| EMR-Merging W/ DARE | 40.58 | 93.23 | 86.52 | 82.86 | 89.93 | 85.98 | 89.68 | 75.09 | $80.48^{+0.30}$ |
| EMR-Merging W/ PAVE | 60.42 | 93.69 | 87.75 | 89.73 | 89.13 | 85.27 | 92.09 | 76.17 | $\mathbf{84.28}^{+4.10}$ |
| Task Arithmetic | 18.78 | 85.89 | 79.90 | 74.03 | 83.78 | 59.08 | 69.67 | 62.09 | 66.65 |
| Task Arithmetic W/ DARE | 18.26 | 85.67 | 80.39 | 73.79 | 83.77 | 59.20 | 69.58 | 61.01 | $66.45^{-0.20}$ |
| Task Arithmetic W/ PAVE | 18.96 | 85.67 | 80.64 | 74.14 | 83.77 | 58.97 | 69.91 | 61.73 | $\mathbf{66.72}^{+0.07}$ |
| Ties-Merging | 20.48 | 84.40 | 81.13 | 58.19 | 85.70 | 64.65 | 74.81 | 42.96 | 64.04 |
| Ties-Merging W/ DARE | 17.56 | 83.83 | 80.39 | 63.60 | 85.79 | 64.82 | 0.7324 | 44.77 | $64.25^{+0.21}$ |
| Ties-Merging W/ PAVE | 25.45 | 82.45 | 79.41 | 64.03 | 85.43 | 69.93 | 77.19 | 51.62 | $\mathbf{66.93}^{+2.89}$ |

Table 2: Results of merging fine-tuned DeBERTa models on eight datasets from GLUE benchmark. The number of examples used to generate context matrix is 4096.

| Methods | CoLA | SST-2 | MRPC | STSB | QQP | MNLI | QNLI | RTE | Average |
|---|---|---|---|---|---|---|---|---|---|
| Individual | 59.10 | 95.06 | 89.21 | 91.31 | 91.50 | 88.52 | 93.30 | 69.31 | 84.66 |
| EMR-Merging | 47.05 | 94.72 | 71.32 | 87.57 | 88.36 | 87.38 | 92.29 | 64.26 | 79.11 |
| EMR-Merging W/ DARE | 47.76 | 94.61 | 72.06 | 87.30 | 88.77 | 87.67 | 92.28 | 66.43 | $79.61^{+0.50}$ |
| EMR-Merging W/ PAVE | 61.32 | 94.50 | 84.31 | 89.54 | 90.34 | 87.50 | 93.01 | 60.65 | $\mathbf{82.64}^{+3.53}$ |
| Task Arithmetic | 3.03 | 75.00 | 68.38 | 52.72 | 63.44 | 43.46 | 51.16 | 50.90 | 51.01 |
| Task Arithmetic W/ DARE | 2.84 | 68.81 | 68.38 | 54.24 | 64.38 | 42.23 | 55.15 | 49.46 | $50.68^{-0.33}$ |
| Task Arithmetic W/ PAVE | 2.18 | 75.46 | 68.38 | 55.90 | 64.00 | 43.48 | 61.25 | 51.99 | $\mathbf{52.83}^{+1.82}$ |
| Ties-Merging | 5.36 | 66.28 | 68.63 | 19.09 | 63.68 | 40.44 | 55.39 | 48.74 | 45.95 |
| Ties-Merging W/ DARE | 6.03 | 56.42 | 68.87 | 0.61 | 66.28 | 42.11 | 54.00 | 51.26 | $43.19^{-2.76}$ |
| Ties-Merging W/ PAVE | 1.39 | 68.35 | 68.63 | 36.76 | 63.12 | 44.01 | 59.36 | 50.90 | $\mathbf{49.06}^{+3.11}$ |

the corresponding task to construct knowledge-aware subspace. When using CoLA samples to generate the covariance matrix, performance drops significantly across all tasks. These results suggest knowledge-awareness is crucial for successful rank preservation, and emphasize the importance of introducing knowledge-aware subspaces for purifying task vectors.

## 4.2 MODEL MERGING FOR GLUE TASKS

To evaluate the effectivness of PAVE, we first present results on GLUE tasks using RoBERTa as the backbone model. The numerical results are summarized in Table 1. We observe that PAVE yields a notable average performance gain of +4.10% when applied to EMR-Merging, reducing the gap to just 1.27% from the individual models' average performance. For specific tasks such as CoLA, PAVE significantly improves the score from 39.96% to 60.42%, even surpassing the individual model's score of 60.18%. These improvements are not limited to EMR-Merging. PAVE also shows improvements with Ties-Merging and Task Arithmetic strategies. For instance, when integrating on Ties-Merging, PAVE achieves an average gain of +2.89%, outperforming both the baseline and the DARE-enhanced variant.

Similar to the experimental setting for RoBERTa, we evaluate our method on DeBERTa and present the results in Table 2. PAVE cooperates effectively with EMR-Merging, achieving a substantial average performance gain of +3.53%, narrowing the average gap to 2.02% compared to the averaged performance (84.66%) of individual models. Notably, on CoLA, PAVE boosts performance to a new high of 61.32%, even surpassing the individual model's score of 59.10%, also achieving an improvement of 14.27% over the EMR-Merging baseline. Beyond EMR-Merging, PAVE also improves results in Task Arithmetic and Ties-Merging settings. In Task Arithmetic, it raises the average performance by +1.82%, while in Ties-Merging, it achieves a considerable gain of +3.11% over the original baseline.

Table 3: Results of merging fine-tuned LLaMA-2-7B models on Math and Code tasks. The number of examples used to generate the context matrix is 512.

| Methods | Mathematical Solving | | Code Generating | |
|---|---|---|---|---|
| | GSM8K | MATH | Human Eval | MBPP |
| Individual Math | 54.9 | 10.7 | - | - |
| Individual Code | - | - | 39.6 | 30.1 |
| Average Merging | 32.2 | 5.0 | 29.2 | 29.0 |
| Average Merging w/ DARE | $31.9^{-0.3}$ | $5.0^{+0.0}$ | $29.8^{+0.6}$ | $28.6^{-0.4}$ |
| Average Merging w/ PAVE | $\mathbf{32.4}^{+0.2}$ | $\mathbf{5.0}^{+0.0}$ | $\mathbf{31.1}^{+1.9}$ | $\mathbf{29.0}^{+0.0}$ |
| Task Arithmetic | 34.3 | 4.1 | 33.5 | 27.6 |
| Task Arithmetic w/ DARE | $32.3^{-2.0}$ | $\mathbf{4.4}^{+0.3}$ | $32.3^{-1.2}$ | $\mathbf{28.6}^{+1.0}$ |
| Task Arithmetic w/ PAVE | $\mathbf{34.5}^{+0.2}$ | $4.2^{+0.1}$ | $\mathbf{35.4}^{+1.9}$ | $28.0^{+0.4}$ |

Table 4: Results of merging fine-tuned DeBERTa models on eight datasets from GLUE benchmark. The number of examples used to generate context matrix is 4096.

| Methods | CoLA | SST-2 | MRPC | STSB | QQP | MNLI | QNLI | RTE | Average |
|---|---|---|---|---|---|---|---|---|---|
| Individual | 59.10 | 95.06 | 89.21 | 91.31 | 91.50 | 88.52 | 93.30 | 69.31 | 84.66 |
| EMR-Merging | 47.05 | 94.72 | 71.32 | 87.57 | 88.36 | 87.38 | 92.29 | 64.26 | 79.11 |
| EMR-Merging W/ PAVE plain | 59.88 | 94.61 | 82.60 | 88.83 | 90.40 | 87.35 | 92.44 | 58.48 | 81.82 |
| EMR-Merging W/ PAVE | 61.32 | 94.50 | 84.31 | 89.54 | 90.34 | 87.50 | 93.01 | 60.65 | **82.64** |
| Task Arithmetic | 3.03 | 75.00 | 68.38 | 52.72 | 63.44 | 43.46 | 51.16 | 50.90 | 51.01 |
| Task Arithmetic W/ PAVE plain | 2.56 | 74.73 | 68.38 | 54.83 | 64.03 | 42.84 | 60.19 | 51.16 | 52.34 |
| Task Arithmetic W/ PAVE | 2.18 | 75.46 | 68.38 | 55.90 | 64.00 | 43.48 | 61.25 | 51.99 | **52.83** |
| Ties-Merging | 5.36 | 66.28 | 68.63 | 19.09 | 63.68 | 40.44 | 55.39 | 48.74 | 45.95 |
| Ties-Merging W/ PAVE plain | 1.04 | 70.07 | 68.87 | 31.33 | 63.16 | 43.89 | 57.79 | 50.18 | 48.29 |
| Ties-Merging W/ PAVE | 1.39 | 68.35 | 68.63 | 36.76 | 63.12 | 44.01 | 59.36 | 50.90 | **49.06** |

## 4.3 MODEL MERGING FOR GENERATIVE TASKS

For generative tasks, we conduct experiments merging the math model WizardMath-7B-V1.0 (Luo et al., 2023) and the code model LLaMA-2-7B-EvolCodeAlpaca (Agarwalla et al., 2024). As shown in Table 3, PAVE demonstrates a marked improvement of +1.9% on Human Eval compared to Average Merging and Task Arithmetic. Notably, PAVE achieves the highest Human Eval score (35.4%) among all merging strategies, indicating its effectiveness of our purified task vectors in code generation tasks. In the mathematical solving tasks, PAVE consistently maintains or modestly improves performance across both GSM8K (+0.2%) and MATH (+0.1%) compared to the Task Arithmetic baseline. It is worth noting that, in contrast to the randomness in DARE, PAVE is more stable and practical, as none of the merged models show any degradation on the individual tasks. This stability, combined with consistent improvements across multiple tasks, highlights the robustness and practical advantage of PAVE.

## 4.4 ABLATION STUDIES

We evaluate our rank allocation strategy Algorithm 1 in Table 4, using the same DeBERTa experiment settings on models and merging methods in Section 4.2. The plain method uses a fixed preserved rank ratio across all layers and models. Compared to this baseline, our adaptive strategy consistently improves performance across different merging settings. In EMR-Merging, the rank strategy improves the average performance from 81.82% to 82.64% (+0.82%). In other merging methods like Task Arithmetic and Ties-Merging, the gains are +0.49% (from 52.34% to 52.83%) and +0.77% (from 48.29% to 49.06%), respectively. These results suggest that uniform pruning represents a naive and suboptimal approach, while our rank allocation strategy enables better performance under the same pruning budget. Additionally, we include further experiments, presenting

numerical results when PAVE is integrated with alternative decomposition methods, as well as an analysis of sample size and runtime performance. Please refer to Appendix B for the results.

## 5 CONCLUSION

In this paper, we propose the purified task vector (PAVE) as a plug-in method for task vector-based merging approaches. PAVE employs context-oriented decomposition to identify and prune redundancies of the task vectors in knowledge-aware subspace, thereby mitigating conflicts arising from redundancy. To further optimize the rank allocation across different fine-tuned models and improve the merging performance, we introduce a spectral rank allocation strategy that optimizes the summation of normalized activated pruning errors. In experiment, we demonstrate the effectiveness of PAVE with a wide range of merging methods, tasks and model architectures. Furthermore, we explore the integration of alternative decomposition techniques within the PAVE framework and confirm its compatibility with existing model merging approaches.

## 6 ETHICS STATEMENT

We affirm that our work complies with the ICLR Code of Ethics. All authors have read and adhered to the Code of Ethics throughout the research, submission, and discussion phases. Our study does not involve human subjects, personally identifiable information, or any data that raises privacy concerns. All datasets used are publicly available and commonly employed in the research community. To the best of our knowledge, the data sources and models presented do not contain or propagate harmful biases, nor do they pose risks of discrimination or unfair treatment against individuals or groups. We also confirm that no sensitive or potentially harmful applications arise from our findings. The methodologies used are standard within the domain and do not introduce novel risks in terms of misuse or security vulnerabilities. No conflicts of interest or ethical concerns related to funding, sponsorship, or institutional pressure have influenced the conduct or presentation of this research. All contributions were made in accordance with principles of transparency, reproducibility, and research integrity.

## 7 REPRODUCIBILITY STATEMENT

We have made extensive efforts to ensure the reproducibility of our results. All experimental settings, models and architectures are described in detail in Section 4 of the main text and further elaborated in Appendix A. All experiments in our paper were conducted with large sample sizes to minimize variance introduced by sampling. We further provide experiments illustrating the effect of sampling in Appendix B.2. Hyperparameters, hardware specifications are reported comprehensively to enable fair and consistent reproduction of our findings.

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

## A  IMPLEMENTATION DETAILS

### A.1  MODEL MERGING ON GLUE BENCHMARK

In Section 4.2, we employ RoBERTa-base (Liu et al., 2019) and DeBERTa-base (He et al., 2020) as the base model. The performance of each method is evaluated by eight tasks in the GLUE benchmark tasks, including CoLA (Warstadt et al., 2019), SST-2 (Socher et al., 2013), MRPC (Dolan & Brockett, 2005), STS-B (Cer et al., 2017), QQP (Iyer et al., 2017), MNLI (Williams et al., 2017), QNLI (Rajpurkar et al., 2016), RTE (Giampiccolo et al., 2007). For evaluation, CoLA is assessed using the Matthews correlation coefficient, STS-B is measured by the average of Pearson and Spearman correlation coefficients, and the remaining tasks are evaluated based on accuracy. The RoBERTa checkpoints used for model merging are publicly available via the links in (Huang et al., 2024). The DeBERTa checkpoints are obtained by training DeBERTa-base individually on each GLUE task for 5 epochs using the AdamW (Loshchilov & Hutter, 2017) optimizer with a learning rate of $2 \times 10^{-5}$, a warm-up phase of 100 steps, and a weight decay of $10^{-2}$.

Following the setting in Huang et al. (2024), we report the numerical experiments of PAVE with various merging methods including Task Arithmetic (Ilharco et al., 2022), Ties-Merging (Yadav et al., 2023) and EMR-Merging (Huang et al., 2024). We also evaluate these methods with plug-in method DARE (Yu et al., 2024). Following the standard setting of each method, the hyperparameter $\lambda$ in Task Arithmetic and Ties-Merging is selected from {0.1, 0.3, 0.5, 0.7 0.9} based on the best performance. Specifically, for the RoBERTa experiments, we set $\lambda = 0.3$ for Task Arithmetic and $\lambda = 0.9$ for Ties-Merging; for BERT experiments, we set $\lambda = 0.1$ and $\lambda = 0.5$, respectively. The dropout ratio for DARE is fixed to 0.1 to avoid instability. The preserved rank ratio $\rho$ of PAVE is chosen from $\{\frac{7}{8}, \frac{15}{16}, \frac{31}{32}, \frac{63}{64}\}$, and the stopping ratio is set to $\gamma = \rho - \frac{1-\rho}{2}$.

To attain optimal performance on GLUE benchmarks, PAVE progressively progressively assigns full rank to $\Delta W_{\text{PAVE}}$ according to the performance of the plain merging method. This design reflects the observation that certain tasks are more sensitive and cannot tolerate aggressive pruning. At each step, we apply full rank to the the task whose performance is furthest from that of its individual model. This process continues until all task vectors are full rank. As a result, we obtain $K$ merged models and select the one with the highest average performance. Since model merging is not computationally expensive and requires no training, the computational cost of such procedure remains acceptable.

### A.2  MODEL MERGING ON GENERATION BENCHMARKS

Following the setup of DARE presented in Yu et al. (2024), we evaluate PAVE on mathematical solving and code generation tasks, as described in Section 4.3. For mathematical solving, we consider GSM8K (Cobbe et al., 2021) and MATH (Hendrycks et al., 2021), while for code generation, we use HumanEval (Chen et al., 2021) and MBPP (Austin et al., 2021). We adopt the LLaMA-2-7B (Touvron et al., 2023) model as the base model. For mathematical solving, we utilize the fine-tuned model WizardMath-7B-V1.0 (Luo et al., 2023), while for code generation, we employ LLaMA-2-7B-EvolCodeAlpaca (Agarwalla et al., 2024). Additionally, code samples are drawn from the CodeFeedBack dataset (Zheng et al., 2024) to support covariance matrix generation. The Task Arithmetic hyperparameter $\lambda$ is selected from the set $\{0.5, 1\}$, and the dropout ratio for DARE is fixed at 0.2. The pruning rank for PAVE is chosen from $\{4, 6, 8, 16\}$.

## B  ADDITIONAL ANALYSIS AND RESULTS

### B.1  TASK VECTOR WITH VARIOUS DECOMPOSITION METHODS

In this section, we present numerical experiments evaluating the performance of various decomposition methods applied to our decomposition-based task vector. The setting of models and merging methods follow the DeBERTa settings in Section 4.2. The evaluated decomposition methods include: The basic SVD method (Meng et al., 2024), directly approximates the fine-tuned weights via a plain SVD decomposition. ASVD (Yuan et al., 2023), incorporates an input-dependent diagonal scaling matrix derived from the average absolute values of the previous layer's outputs. SVD-LLM (Wang et al., 2024b), applies Cholesky decomposition to the previous layer's output.

Table 5: Results of merging fine-tuned DeBERTa models on eight datasets from GLUE benchmark. The number of examples used to generate context matrix is 4096.

| Methods | CoLA | SST-2 | MRPC | STSB | QQP | MNLI | QNLI | RTE | Average |
|---|---|---|---|---|---|---|---|---|---|
| Individual | 60.18 | 94.04 | 89.22 | 90.63 | 91.41 | 87.20 | 92.71 | 79.06 | 85.55 |
| EMR-Merging | 47.05 | 94.72 | 71.32 | 87.57 | 88.36 | 87.38 | 92.29 | 64.26 | 79.11 |
| EMR-Merging W/ SVD | 62.20 | 93.58 | 83.82 | 89.40 | 89.45 | 76.72 | 92.53 | 58.48 | 80.77 |
| EMR-Merging W/ ASVD | 62.33 | 95.07 | 84.56 | 89.73 | 89.98 | 87.44 | 92.86 | 61.01 | **82.87** |
| EMR-Merging W/ SVD-LLM | 61.82 | 94.72 | 84.07 | 89.55 | 90.10 | 87.44 | 92.90 | 61.37 | 82.74 |
| EMR-Merging W/ PAVE | 61.32 | 94.50 | 84.31 | 89.54 | 90.34 | 87.50 | 93.01 | 60.65 | 82.64 |
| Task Arithmetic | 3.03 | 75.00 | 68.38 | 52.72 | 63.44 | 43.46 | 51.16 | 50.90 | 51.01 |
| Task Arithmetic W/ SVD | 2.96 | 70.30 | 68.38 | 37.90 | 64.30 | 42.88 | 67.01 | 53.07 | 50.85 |
| Task Arithmetic W/ ASVD | 2.53 | 76.03 | 66.38 | 55.60 | 64.08 | 43.23 | 61.27 | 51.62 | 52.59 |
| Task Arithmetic W/ SVD-LLM | 2.74 | 74.89 | 68.38 | 55.48 | 64.02 | 43.54 | 61.45 | 51.62 | 52.76 |
| Task Arithmetic W/ PAVE | 2.18 | 75.46 | 68.38 | 55.90 | 64.00 | 43.48 | 61.25 | 51.99 | **52.83** |
| Ties-Merging | 5.36 | 66.28 | 68.63 | 19.09 | 63.68 | 40.44 | 55.39 | 48.74 | 45.95 |
| Ties-Merging W/ SVD | -3.29 | 53.90 | 68.38 | 6.15 | 62.98 | 35.34 | 54.22 | 52.71 | 41.29 |
| Ties-Merging W/ ASVD | -1.74 | 66.86 | 68.63 | 17.19 | 62.84 | 40.33 | 53.10 | 46.57 | 44.22 |
| Ties-Merging W/ SVD-LLM | 1.13 | 67.89 | 68.63 | 23.33 | 63.09 | 43.02 | 57.84 | 48.74 | 46.70 |
| Ties-Merging W/ PAVE | 1.39 | 68.35 | 68.63 | 36.76 | 63.12 | 44.01 | 59.36 | 50.90 | **49.06** |

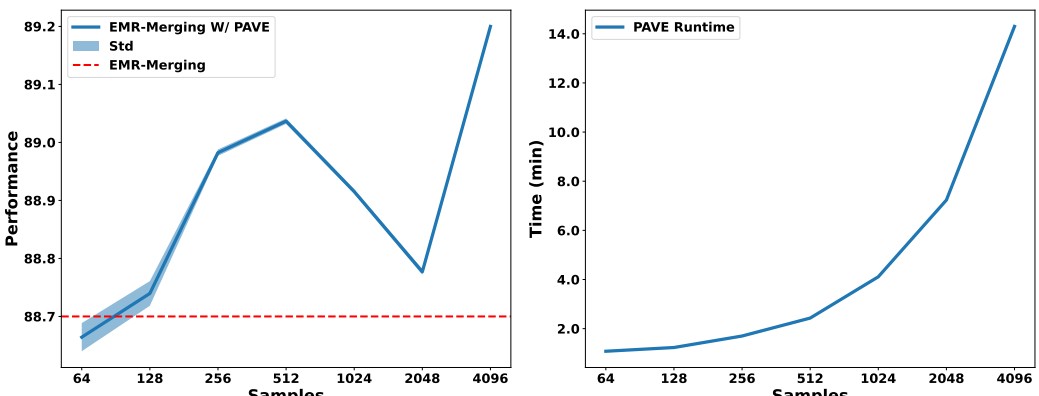

Figure 4: Performance and runtime analysis of PAVE across different sample sizes. **Left:** Mean performance (solid line) and standard deviation (shaded region) of EMR-Merging with PAVE, showing consistent improvements as more samples are used. **Right:** PAVE runtime grows with number of samples.

CO-SVD (Yang et al., 2024b), as the default decomposition method embedded in PAVE, utilizes the covariance matrix of the output of the previous layer as the activation matrix. Among the evaluated methods, CO-SVD demonstrates the most consistent performance on Task Arithmetic and Ties-Merging, while also exhibiting performance comparable to EMR-Merging. These results highlight the strong potential of our decomposition-based purified task vector framework, which is designed to integrate seamlessly with a variety of decomposition methods.

### B.2 ANALYSIS OF SAMPLE SIZE AND TIME EFFICIENCY

We next analyze the impact of sample size on the stability, efficiency, and reproducibility of PAVE. As illustrated in Figure 4, increasing the sample size consistently reduces the variance and improves the overall performance of EMR-Merging with PAVE, highlighting its stability and robustness. When the sample size is very small, due to the regularization, the method degenerates to plain SVD, which explains the lower performance observed in this regime. Importantly, we observe that

Table 6: Results of merging fine-tuned ViT models on 8 image classification benchmarks. The number of samples used to generate context matrix is 4096.

| Methods | SUN397 | Cars | RESISC45 | EuroSAT | SVHN | GTSRB | MNIST | DTD | Average |
|---|---|---|---|---|---|---|---|---|---|
| Individual ViT-B/32 | 75.3 | 77.7 | 96.1 | 99.7 | 97.5 | 98.7 | 99.7 | 79.4 | 90.5 |
| EMR-Merging | 75.2 | 72.8 | 93.5 | 99.5 | 96.9 | 98.1 | 99.6 | 74.4 | 88.7 |
| EMR-Merging W/PAVE | 76.3 | 74.3 | 93.8 | 99.3 | 96.7 | 97.6 | 99.6 | 76.1 | **89.2**$^{+0.5}$ |
| Individual ViT-B/16 | 79.0 | 87.0 | 96.4 | 99.1 | 97.7 | 99.0 | 99.7 | 82.3 | 92.5 |
| EMR-Merging | 78.5 | 82.6 | 95.4 | 99.1 | 97.6 | 98.8 | 99.6 | 78.3 | 91.2 |
| EMR-Merging W/PAVE | 80.2 | 82.8 | 96.1 | 99.2 | 97.6 | 98.7 | 99.6 | 78.9 | **91.7**$^{+0.5}$ |
| Individual ViT-L/14 | 82.3 | 92.4 | 97.4 | 100 | 98.1 | 99.2 | 99.7 | 84.1 | 94.2 |
| EMR-Merging | 83.1 | 90.7 | 96.7 | 99.7 | 97.9 | 99.1 | 99.6 | 82.7 | 93.7 |
| EMR-Merging W/PAVE | 83.6 | 91.7 | 97.2 | 99.8 | 99.7 | 98.9 | 99.7 | 82.2 | **93.9**$^{+0.2}$ |

even a moderate sample size (256–512) is sufficient to yield substantial performance gains, offering a practical trade-off between accuracy and computational cost. This finding suggests that PAVE can be effectively deployed in practice without incurring excessive runtime overhead, while still preserving most of the performance benefits of using larger sample sizes. Throughout our numerical experiments, we increased the number of samples up to 4096 ensure a stable and consistent demonstration of our method's performance. Employing larger sample sizes also helps ensure the reproducibility of our results. All runtime measurements were conducted on a workstation equipped with a single NVIDIA A100 GPU and 8 CPUs, with 512 GB of RAM.

### B.3 MODEL MERGING FOR VIT MODELS

**Settings.** Following the setup of EMR-Merging presented in Huang et al. (2024), we employ ViT-B/32, ViT-B/16 and ViT-L/14 (Radford et al., 2021) as the base model to evaluate the performance of PAVE across 8 image classification tasks: SUN397 (Xiao et al., 2010), Cars Krause et al. (2013), RESISC45 (Cheng et al., 2017), EuroSAT (Helber et al., 2019), SVHN (Netzer et al., 2011), GT-SRB (Stallkamp et al., 2011), MNIST (LeCun, 1998), and DTD (Cimpoi et al., 2014). The preserved rank ratio $\rho$ of PAVE is chosen from $\{\frac{7}{8}, \frac{15}{16}, \frac{31}{32}, \frac{63}{64}\}$, and the stopping ratio is set to $\gamma = \rho - \frac{1-\rho}{2}$. The checkpoints are publicly accessible via the link provided in Ilharco et al. (2022).

**Results.** As shown in Table 6, while the plain EMR-Merging method already demonstrates strong performance, PAVE still consistently enhances the performance across all ViT variants. For ViT-B/32, PAVE improves the average accuracy by +0.5% (from 88.7% to 89.2). On the more capable ViT-B/16 model, PAVE again contributes an improvement of +0.5%, increasing the average from 91.2% to 91.7%.

## C THE USE OF LARGE LANGUAGE MODELS

We understand and follow the policies regarding the use of large language models (LLMs). In this work, LLMs were used only to aid or polish the writing, mainly for fixing grammar issues and improving clarity. They were not involved in generating content, developing ideas, or conducting research. The authors take full responsibility for all parts of the manuscript.

