# OpenReview forum: "Purifying Task Vectors in Knowledge-Aware Subspace for Model Merging"
_ICLR.cc/2026/Conference — Submitted to ICLR 2026_

### Official Review · Reviewer_xa9N · 2025-10-27

**Soundness:** 2
**Presentation:** 2
**Contribution:** 2
**Rating:** 4
**Confidence:** 4

**Summary:**

This paper proposed purifying task vectors (PAVE) to eliminate the task-irrelevant redundancy in task vectors, aiming to address performance degradation in model merging. As previous methods rely on randomness and lack knowledge awareness, PAVE utilizes data to filter out the task-relevant and redundant subspaces in parameters. They further proposed a rank allocation strategy to adaptively assign ranks to models. The experiments on diverse models and datasets show the effectiveness of their method.

**Strengths:**

- The motivation of this paper is novel: as previous methods focus on removing the conflicts in the whole task vectors, this paper proposes to eliminate noise and redundancy in the subspaces.
- The experiments are comprehensive in terms of models and datasets.
- The empirical results show the effectiveness of the main proposed components in their methods, including the purification and rank allocation.

**Weaknesses:**

- This method requires access to data from the target tasks, where the models come from. However, this assumption may not be valid in some cases (e.g., using HuggingFace models or when the data is private) and limits the application scenarios of this method.
- If my understanding is correct, the motivation of Sec.3.2 is to reduce the task conflicts. However, it is unclear to me why the proposed method in Sec.3.2 can help reduce the task conflicts. I think it would be better to clarify further the connection between the motivation and the proposed method.
- The motivation to use spectral rank allocation is not clear. Please see the questions.
- Though the performance is improved, PAVE introduces more computational cost. The trade-off between performance and computing should be studied to better justify the benefit of PAVE. Also, such a trade-off of using the rank allocation strategy should be studied in Tab.4, where the performance improvement is so limited.
- It seems that ThanoRA [1] proposed a similar method, but it is not discussed in this paper.

[1] Liang, Jian, et al. "ThanoRA: Task Heterogeneity-Aware Multi-Task Low-Rank Adaptation." arXiv preprint arXiv:2505.18640 (2025).

**Questions:**

- In lines 262-263, why is assigning the rank $r$ to each model non-trivial? I think it would be better to clarify the reason.
- As the author assumes the data access, I think it is better to evaluate the performance on some data-dependent merging algorithms, such as Fisher merging and RegMean.
- In Tab.3, the performance improvement is not as significant as that in Tab.1 and 2. Does it mean that as the model scale increases, the impact of PAVE becomes less? If PAVE cannot perform well with the scaling law, its contribution could be limited, as larger models are more necessary to be merged.

---

> ### Author Response · Authors · 2025-11-21
> **Official Comment by Authors (Part1)**
>
> Thank you for recognizing the contributions of our work and for your valuable feedback. Below, we address each of the identified weaknesses and questions in detail.
>
> **Weakness 1: Accessing data concerns**
>
> We understand your concerns. However, data access is actually allowable in the field of model merging. Some influential merging methods like Fisher-Merging [1], RegMean-Merging [2], and AdaMerging [3] also require data from the tasks to help with merging, evidence can be found in Equation 2 in [1], Equation 1 in [2] and Section 3.2.2 in [3]. And for the condition that original training data is strictly private or inaccessible, we provide two alternative solutions for it, please refer to the global comments for more details and experimental results.
>
> **Weakness 2 : Clarify how PAVE help reducing conflicts**
>
> PAVE reduces merging conflicts by removing redundancy components in the plain task vectors. The underlying motivation for PAVE is based on the following insight. Recent studies, such as DARE [4], Subspace Boosting [5], TSV-Merging [6], and WUDI-Merging [7], have observed that task vectors often contain significant redundancy, meaning that many parameter updates include noise or information that isn’t relevant to the task. When merging several fine-tuned models, these redundant components tend to accumulate and interfere, leading to performance degradation in the merged model. To mitigate this issue, recent approaches like DARE have attempted to reduce conflicts by randomly masking task vectors using Dropout to induce sparsity and reduce conflict. However, the Dropout operation in DARE involves randomness and lacks knowledge awareness, and thus it cannot locate the valuable components in a targeted manner. To overcome this limitation, inspired by [8,9,10], we notice that task-specific abilities actually lie in the subspace of pre-trained parameters. Therefore, redundant components in the task vector should be identified and removed in subpace. Following this motivation, we propose purified task vectors. First, we construct knowledge-aware subspaces by applying CO-SVD to the purified task vector; then, we identify and remove the redundancy in knowledge-aware subspaces to reduce merging conflicts. A visual illustration of this process is provided in Figure 1.
>
>
>
> **Weakness 3 & Question 1:  Why the r allocation is non-trivial, clarify the rank strategy and explain**
>
>
> Allocating rank r to each model is non-trivial mainly because the fine-tuned models vary in their adaptation to rank pruning; some fine-tuned weights are more robust and can preserve performance even after aggressive rank pruning so need less ranks, while others are more sensitive and require more ranks. As a result, a naive rank assignment strategy proves suboptimal, as shown numerically in Table 4 in our paper, applying the rank allocation strategy will further improve the overall performance.
>
> This strategy determines the pruning ranks for each model by optimizing Equation 9 using a greedy via a greedy algorithm, detailed in Algorithm 1. The Equation 9 is built on a term named activate pruning error, defined in Equation 7. This error estimates the activated distance between our purified task vector and the original plain task vector. Since the plain task vector retains the full capabilities through fine-tuning, this error serves as a measure of how well the purified task vector preserves task-specific knowledge. The optimization objective in Equation 9 minimizes a normalized sum of these pruning errors across all models.
>
> More details of rank allocation strategy are presented in Section 3.3, we also updated more content to further explain the motivation of the rank strategy.

---

> > ### Author Response · Authors · 2025-11-21
> > **Official Comment by Authors (Part2)**
> >
> > **Weakness 4: Discussion on computational time**
> >
> > The trade-off between performance and computational cost is illustrated in Figure 4, where the subfigure on the left shows the performance over number of samples, figure on the right studies computation time with respect to the number of samples. The computational cost of the rank allocation strategy is light, as the key component is a greedy algorithm. The most computational intensive step in PAVE is SVD. We also conduct an experiment by merging eight ViT-B-16 models to validate our conclusion. In this experiment, the decomposition took 146 seconds, while the greedy algorithm for rank allocation took just 1 second.
> >
> > Although the performance improvement from the rank strategy is limited, the cost is negligible, allowing us to obtain additional gains at merely no cost.
> >
> > **Weakness 5: ThanoRA paper**
> >
> > Thanks for bringing the ThanoRA paper to us, we have cited it and added it to our updated related work section. We understand both our method and the method in ThanoRA utilize the same decomposition method (CO-SVD).
> > However, ThanoRA is designed for low-rank adaptation, where trainable adapters are introduced and optimized for learning downstream tasks.
> > In contrast, our method, PAVE, focuses on model merging and is training-free. Therefore, the task and goal of ThanoRA are fundamentally different from model merging, and we cannot compare the experiment performance with ThanoRA.

---

> > > ### Author Response · Authors · 2025-11-21
> > > **Official Comment by Authors (Part3)**
> > >
> > > **Question 2: Comparison with data-dependent merging algorithms**
> > >
> > > Data-dependent algorithms like RegMean-Merging and Fisher-Merging are not based on task vectors. As PAVE is a plug-in method designed for task-vector based merging methods, we cannot integarte PAVE on RegMean-Merging and Fisher-Merging. So, we directly compare our performance with Regmean-merging and Fisher-Merging as follows:
> > >
> > >
> > > | ViT-B-32              | SUN397 | Cars  | RESISC45 | EuroSAT | SVHN | GTRSB | MNIST | DTD  | Average |
> > > |-----------------------|--------|-------|----------|---------|------|-------|-------|------|---------|
> > > | Individual            | 75.3   | 77.7  | 96.1     | 99.7    | 97.5 | 98.7  | 99.7  | 79.4 | 90.5    |
> > > | Fisher-Merging        | 68.6   | 69.2  | 70.7     | 66.4    | 72.9 | 51.1  | 87.9  | 59.9 | 68.3    |
> > > | RegMean-Merging       | 65.3   | 63.5  | 75.6     | 78.6    | 78.1 | 67.4  | 93.7  | 52.0 | 71.8    |
> > > | EMR-Merging           | 75.2   | 72.8  | 93.5     | 99.5    | 96.9 | 98.1  | 99.6  | 74.4 | 88.7    |
> > > |  EMR-Merging w/ PAVE  | 76.3   | 74.3  | 93.8     | 99.3    | 96.7 | 97.6  | 99.6  | 76.1 | 89.2    |
> > >
> > > In conclusion, our method, when combined with EMR-Merging, consistently achieves superior performance among the other data-dependent merging approaches.
> > >
> > >
> > > **Question 3: Performance on scaled-up models**
> > >
> > > The performance in Table 3 is not as significant as Table 1 and 2 are primarily due to the increased difficulty of merging large-scale models and complexity of the evaluation tasks, like mathematical reasoning and code generation. This challenge is well-recognized in the model merging area. Most numerical experiments in prior merging work focus **only** on relatively small models such as RoBERTa or ViT, for example TSV-Merging, ISO-Merging [11], and plug-in approaches like AWD-Merging [12]. Even when generative tasks are included, as in WUDI-Merging, the performance is still unsatisfactory (e.g., the Table 6 in wudi-merging paper, the average performance of merging Qwen-14B only has 0.99% increase compared to the baseline). Moreover, the current plug-in method, DARE, is facing performance degradation in our evaluation.
> > >
> > > We quote Table 6 results in WUDI-Merging here:
> > >
> > > | Qwen-14B              | MMLU  | TruthfulQA | BBQ    | CNN   | Average |
> > > |-----------------------|-------|------------|--------|-------|---------|
> > > | Individual            | 68.35 | 53.34      | 93.53  | 19.46 | 58.67   |
> > > | Task Arithmetic       | 67.56 | 52.33      | 78.38  | 20.54 | 54.7    |
> > > | WUDI-Merging          | 69.17 | 55.71      | 80.56  | 17.33 | 55.69   |
> > >
> > >
> > > We quote Table 3 in our paper:
> > >
> > > | Llama-2-7B                 | GSM8K | MATH | HuamEval | MBPP | Average |
> > > |---------------------------|-------|------|----------|------|---------|
> > > | Individual                | 54.9  | 10.7 | 39.6     | 30.1 | 33.8    |
> > > | Task Arithmetic           | 34.3  | 4.1  | 33.5     | 27.6 | 24.8    |
> > > | Task Arithmetic w/ DARE   | 32.3  | 4.4  | 32.3     | 28.6 | 24.4    |
> > > | Task Arithmetic w/ PAVE   | 34.5  | 4.2  | 35.4     | 28   | 25.5    |
> > >
> > >
> > >
> > > In contrast, our method achieves more stable improvement than DARE and does not suffer from such degradation. Therefore, although the improvement of our method on generative tasks is not as large as those in the understanding tasks, our performance is actually an advancement of this field.
> > >
> > >
> > > [1] Merging Models with Fisher-Weighted Averaging
> > >
> > > [2] Dataless Knowledge Fusion by Merging Weights of Language Models
> > >
> > > [3] RegMean++: Enhancing Effectiveness and Generalization of Regression Mean for Model Merging
> > >
> > > [4] Language Models are Super Mario: Absorbing Abilities from Homologous Models as a Free Lunch
> > >
> > > [5] Subspace-boosted Model Merging
> > >
> > > [6] Task Singular Vectors: Reducing Task Interference in Model Merging
> > >
> > > [7] Whoever Started the Interference Should End It: Guiding Data-Free Model Merging via Task Vectors
> > >
> > > [8] CorDA: Context-Oriented Decomposition Adaptation of Large Language Models for Task-Aware Parameter-Efficient Fine-tuning
> > >
> > > [9] SVCCA: Singular Vector Canonical Correlation Analysis for Deep Learning Dynamics and Interpretability
> > >
> > > [10] Measuring the Intrinsic Dimension of Objective Landscapes
> > >
> > > [11] No Task Left Behind: Isotropic Model Merging with Common and Task-Specific Subspaces
> > >
> > > [12] Multi-Task Model Merging via Adaptive Weight Disentanglement

---

> > ### Author Response · Authors · 2025-11-26
> > **Thanks again**
> >
> > Dear Reviewer xa9N:
> >
> > We would like to thank you again for your time and helpful feedback.
> >
> > We have tried our best to address all your comments in our response and have updated the paper accordingly. We hope you will consider raising your score based on our revisions. We would appreciate any chance to continue the discussion. Please feel free to let us know if you have any further questions or remaining concerns.
> >
> > Best,
> >
> > Authors

---

> > > ### Comment · Reviewer_xa9N · 2025-11-26
> > >
> > > Thanks for your rebuttal. Here are my remaining questions:
> > >
> > > **W2**: Is Eq.5 equivalent to use SVD to decompose $\Delta W = W_{FT} - W_B$? The motivation for using CO-SVD is that pretrained models have mixed knowledge; how about using the learned $\Delta W$, which is supposed to have task-specific knowledge?
> > >
> > > **W4**: In Fig.4 (left), the line should be monotonically non-decreasing. Why is there an inflection point at 2048? Is it due to some (random) variance since the range is so small (0.5%)? I think more examples would always give more "context" to help CO-SVD. Also, since larger models are more difficult to (re-)train, it is more necessary to achieve better performance on larger models. But in Fig.3, the improvement in Llama models is obviously smaller. Does it mean that PAVE lacks benefit when using larger models?
> > >
> > > **W5**: Good to know that. Since ThanoRA is different from this paper's scope, it is not necessary to cite and discuss it.
> > >
> > > **Q2**: I think the main improvement comes from EMR-merging. It is an adaptive merging method, so of course, it is better than those unified merging methods.
> > >
> > > **Q3**: Sorry but I cannot agree with the author. Wudi-merging has an improvement of 1% improvement while PAVE has 1.1%. Wudi-merging is even evaluated on a larger model with more tasks. Though IMO Wudi-merging's improvement is not significant as well, I do not see that PAVE is more stable than Wudi.

---

> > > > ### Author Response · Authors · 2025-11-28
> > > > **Reply to Reviewer xa9N's Questions (Part 2)**
> > > >
> > > > To the W4 large scale model performance concerns
> > > >
> > > > We agree that achieving better performance on larger models is more important. However, the challenges associated with merging large-scale models and evaluating on generative tasks are well-recognized in the mdoel merging literature. Most numerical experiments in prior merging work focus only on relatively small models such as RoBERTa or ViT, for example TSV-Merging, ISO-Merging, and plug-in approaches like AWD-Merging. Even when generative tasks are considered, for example, in Table 6 of WUDI-Merging, the observed improvements remain limited.
> > > >
> > > > We quote Table 6 results in WUDI-Merging here:
> > > >
> > > > | Qwen-14B              | MMLU  | TruthfulQA | BBQ    | CNN   | Average |
> > > > |-----------------------|-------|------------|--------|-------|---------|
> > > > | Individual            | 68.35 | 53.34      | 93.53  | 19.46 | 58.67   |
> > > > | Task Arithmetic       | 67.56 | 52.33      | 78.38  | 20.54 | 54.7    |
> > > > | WUDI-Merging          | 69.17 | 55.71      | 80.56  | 17.33 | 55.69   |
> > > >
> > > > These results show that even the latest SOTA methods struggle to deliver consistent gains when merging large models for generative tasks.
> > > >
> > > > We quote Table 3 in our paper:
> > > >
> > > > | Llama-2-7B                 | GSM8K | MATH | HuamEval | MBPP | Average |
> > > > |---------------------------|-------|------|----------|------|---------|
> > > > | Individual                | 54.9  | 10.7 | 39.6     | 30.1 | 33.8    |
> > > > | Task Arithmetic           | 34.3  | 4.1  | 33.5     | 27.6 | 24.8    |
> > > > | Task Arithmetic w/ DARE   | 32.3  | 4.4  | 32.3     | 28.6 | 24.4    |
> > > > | Task Arithmetic w/ PAVE   | 34.5  | 4.2  | 35.4     | 28   | 25.5    |
> > > >
> > > > We quote Table 3 in our paper here to highlight that, DARE, as a current plug-in method, is facing performance degradation on subtasks (also see Table 1 in DARE's paper). In contrast, our method PAVE, also serve as a plug-in method, achieve steady improvements across all subtasks when compared to both the baseline method and DARE. This indicates that even under the challenging setting of merging large scale models for generative tasks, our method still shows better results than recent plug-in methods.
> > > >
> > > > Therefore, our work advances the field not only in small model merging for understanding tasks but also in large-scale model merging for generative tasks. It would be unfair to dismiss our contribution solely based on the absolute improvement numbers in large-scale generative settings.

---

> > > > ### Author Response · Authors · 2025-11-28
> > > > **Reply to Reviewer xa9N's Questions (Part 3)**
> > > >
> > > > To the Q2 in your comment:
> > > >
> > > > We understand using EMR-Merging is dynamic, and we have also evaluated PAVE on static merging methods such as Task Arithmetic, Ties-Merging, and recent SOTA WUDI-Merging. These results show that PAVE can still provide consistent improvements in the static setting.
> > > >
> > > > We quote the Table 2 in our paper here:
> > > >
> > > > | DeBERTa                     | CoLA  | SST-2 | MRPC  | STSB  | QQP   | MNLI  | QNLI  | RTE   | Average |
> > > > |-----------------------------|-------|-------|-------|-------|-------|-------|-------|-------|---------|
> > > > | Individual              | 59.10 | 95.06 | 89.21 | 91.31 | 91.50 | 88.52 | 93.30 | 69.31 | 84.66   |
> > > > | EMR-Merging           | 47.05 | 94.72 | 71.32 | 87.57 | 88.36 | 87.38 | 92.29 | 64.26 | 79.11   |
> > > > | EMR-Merging w/ DARE         | 47.76 | 94.61 | 72.06 | 87.30 | 88.77 | 87.67 | 92.28 | 66.43 | 79.61 *(+0.50)* |
> > > > | **EMR-Merging w/ PAVE**    | 61.32 | 94.50 | 84.31 | 89.54 | 90.34 | 87.50 | 93.01 | 60.65 | **82.64 *(+3.53)*** |
> > > > | Task Arithmeti         | 3.03  | 75.00 | 68.38 | 52.72 | 63.44 | 43.46 | 51.16 | 50.90 | 51.01   |
> > > > | Task Arithmetic w/ DARE     | 2.84  | 68.81 | 68.38 | 54.24 | 64.38 | 42.23 | 55.15 | 49.46 | 50.68 *(-0.33)* |
> > > > | **Task Arithmetic w/ PAVE** | 2.18  | 75.46 | 68.38 | 55.90 | 64.00 | 43.48 | 61.25 | 51.99 | **52.83 *(+1.82)*** |
> > > > | Ties-Merging            | 5.36  | 66.28 | 68.63 | 19.09 | 63.68 | 40.44 | 55.39 | 48.74 | 45.95   |
> > > > | Ties-Merging w/ DARE        | 6.03  | 56.42 | 68.87 | 0.61  | 66.28 | 42.11 | 54.00 | 51.26 | 43.19 *(-2.76)* |
> > > > | **Ties-Merging w/ PAVE**    | 1.39  | 68.35 | 68.63 | 36.76 | 63.12 | 44.01 | 59.36 | 50.90 | **49.06 *(+3.11)*** |
> > > >
> > > > We quote the results in the discussion with other reviewer here:
> > > >
> > > > | ViT-B-32               | SUN397 | Cars  | RESISC45 | EuroSAT | SVHN  | GTRSB | MNIST | DTD  | Average |
> > > > |------------------------|--------|-------|----------|---------|-------|-------|-------|------|---------|
> > > > | Individual             | 75.3   | 77.7  | 96.1     | 99.7    | 97.5  | 98.7  | 99.7  | 79.4 | 90.5    |
> > > > | WUDI-Merging           | 69.8  | 68.4  | 82.8    | 94.0   | 90.5 | 93.7 | 99.2 | 68.7| 83.4   |
> > > > |  **WUDI-Merging w/ PAVE**   | 70.5  | 69.7 | 86.0    | 95.4 | 94.0 | 95.0 | 99.3 | 70.0   | **85.0 *(+1.6)***     |
> > > > | EMR-Merging           | 75.2   | 72.8  | 93.5     | 99.5    | 96.9 | 98.1  | 99.6  | 74.4 | 88.7    |
> > > > |  **EMR-Merging w/ PAVE**  | 76.3   | 74.3  | 93.8     | 99.3    | 96.7 | 97.6  | 99.6  | 76.1 | 89.2*(+0.5)***     |
> > > >
> > > >
> > > > Our method, PAVE, serves as a plug-and-play approach. Even though EMR-Merging is a dynamic merging method, we have also compared our approach on static and unified merging baselines, such as Task Arithmetic, Ties-Merging, and WUDI-Merging. This demonstrates that our method is general and achieves consistent improvements across both static and dynamic merging settings.

---

> > > > ### Author Response · Authors · 2025-11-28
> > > > **Reply to Reviewer xa9N's Questions (Part 4)**
> > > >
> > > > To the Q3 in your comment:
> > > >
> > > > We first clarify that merging methods like WUDI-Merging are standalone merging approaches. In contrast, our method, PAVE, serves as a plug-in component, designed as a plug-and-play block to enhance existing merging methods by improving performance. Our method PAVE can be applied to a broad range of merging techniques.
> > > >
> > > > The results from Table 6 in WUDI-Merging and Table 3 in our paper (quoted earlier) primarily aim to demonstrate the challenges of merging large-scale models on generative tasks. In WUDI-Merging, the performance gain on Qwen-14B is limited to about 1%, while DARE shows performance drops on GSM8K and HumanEval when merging Llama-2-7B models. However, the numerical results from these two tables should not be directly compared, apologies for the earlier confusion, because the task setups differ. First, WUDI-Merging evaluates model merging on relatively simpler tasks, combining models fine-tuned on similar knowledge domains (e.g., general knowledge for MMLU, and text generation for CNN). In contrast, our experiments involve more challenging settings such as mathematical reasoning and code generation. Second, the method categories differ: PAVE is a plug-in method, while WUDI-Merging is a merging baseline. A fairer comparison would involve evaluating combinations like WUDI-Merging + PAVE and EMR-Merging + PAVE. We do provide such numerical results on ViT-B-16, which are listed below
> > > >
> > > > We quote the results in the discussion with other reviewer here:
> > > >
> > > >
> > > > | ViT-B-32               | SUN397 | Cars  | RESISC45 | EuroSAT | SVHN  | GTRSB | MNIST | DTD  | Average |
> > > > |------------------------|--------|-------|----------|---------|-------|-------|-------|------|---------|
> > > > | Individual             | 75.3   | 77.7  | 96.1     | 99.7    | 97.5  | 98.7  | 99.7  | 79.4 | 90.5    |
> > > > | WUDI-Merging           | 69.8  | 68.4  | 82.8    | 94.0   | 90.5 | 93.7 | 99.2 | 68.7| 83.4   |
> > > > |  **WUDI-Merging w/ PAVE**   | 70.5  | 69.7 | 86.0    | 95.4 | 94.0 | 95.0 | 99.3 | 70.0   | **85.0 *(+1.6)***     |
> > > > | EMR-Merging           | 75.2   | 72.8  | 93.5     | 99.5    | 96.9 | 98.1  | 99.6  | 74.4 | 88.7    |
> > > > |  **EMR-Merging w/ PAVE**  | 76.3   | 74.3  | 93.8     | 99.3    | 96.7 | 97.6  | 99.6  | 76.1 | **89.2 *(+0.5)***     |
> > > >
> > > >
> > > > From our numerical experiments, we conclude that WUDI-Merging + PAVE achieves an average performance improvement of +1.6 compared to plain WUDI-Merging, while EMR-Merging + PAVE yields a +0.5 improvement and closely approaches the average performance of the individual models.

---

> ### Author Response · Authors · 2025-11-28
> **Reply to Reviewer xa9N's Questions (Part 1)**
>
> Thank you for your reply. We understand that you still have some concerns, and we would like to address them with the following response.
>
> To the W2 in your comment:
>
> No, the Equation 5 is different from directly applying SVD to decompose the task vector. Equation 5 represents our use of CO-SVD and redundant component pruning applied to the fine-tuned weights. This is expressed as $W\_{\text{FT}}^\dagger = \text{SVD}\_r(W\_{\text{FT}}C)C^{-1}$, and the full purified task vector can be shown in $\Delta W\_{\text{PAVE}} = W\_{\text{FT}}^\dagger - W\_{\text{B}} = \text{SVD}\_{r}(W_{\text{FT}}C)C^{-1} - W\_{\text{B}}$. In contrast, directly applying SVD to the task vector yields $\text{SVD}(W_{\text{FT}} - W_{\text{B}})$. Compared to the direct approach, our method is knowledge-aware, as it incorporates a covariance matrix derived from sampled data into the decomposition to guide weight activation. As a result, the components corresponding to large singular values are highly task-relevant, while those associated with small singular values are considered less relevant and redundant.
>
> The motivation for using CO-SVD is that fine-tuned models contain a mixture of pre-existing and task-specific knowledge. Our objective is to identify the components most relevant to the fine-tuning task—specifically, those that are absent from the pre-trained model. Using the learned $\Delta W$ corresponds to the original task vector approach. However, as discussed in our previous response, task vectors often contain significant redundancy. Our method is designed to isolate and eliminate these redundant components, thereby refining the task-specific signal.
>
>
> To the W4 Figure 4 inflection point concerns in your comment:
>
> Thank you for the interesting question! Yes, by intuition, the the line should be monotonically non-decreasing. However, in practice, to guarantee the covariance matrix is invertible, we need to introduce a regularization term $\lambda I$, so the regularized covariance matrix will be  $C+\lambda I$, where $I$ is an identity matrix. The regularization coefficient $\lambda$ is determined to make sure $\|\|(C+\lambda I)(C+\lambda I)^{-1} - I\|\| \leq \epsilon$, where $\epsilon$ constant introduced to account for tolerance threshold during computation. We believe that the inflection point observed at 2048 arises from a mismatch between $\epsilon$ and the sample size, causing instability in regularization at that point. Also, in fact, the drop is very small, only around 0.2 to 0.3 from 512 to 2048, likely due to random fluctuations, which are difficult to avoid as the performance is already near saturated performance levels.
> Besides the mismatch in $\epsilon$, the randomness in data sampling may also contribute to the fluctuation.

---

> ### Author Response · Authors · 2025-11-28
> **Reply to Reviewer xa9N's Questions (Part 3)**
>
> To the Q2 in your comment:
>
> We understand using EMR-Merging is dynamic, and we have also evaluated PAVE on static merging methods such as Task Arithmetic, Ties-Merging, and recent SOTA WUDI-Merging. These results show that PAVE can still provide consistent improvements in the static setting.
>
> We quote the Table 2 in our paper here:
>
> | DeBERTa                     | CoLA  | SST-2 | MRPC  | STSB  | QQP   | MNLI  | QNLI  | RTE   | Average |
> |-----------------------------|-------|-------|-------|-------|-------|-------|-------|-------|---------|
> | Individual              | 59.10 | 95.06 | 89.21 | 91.31 | 91.50 | 88.52 | 93.30 | 69.31 | 84.66   |
> | EMR-Merging           | 47.05 | 94.72 | 71.32 | 87.57 | 88.36 | 87.38 | 92.29 | 64.26 | 79.11   |
> | EMR-Merging w/ DARE         | 47.76 | 94.61 | 72.06 | 87.30 | 88.77 | 87.67 | 92.28 | 66.43 | 79.61 *(+0.50)* |
> | **EMR-Merging w/ PAVE**    | 61.32 | 94.50 | 84.31 | 89.54 | 90.34 | 87.50 | 93.01 | 60.65 | **82.64 *(+3.53)*** |
> | Task Arithmeti         | 3.03  | 75.00 | 68.38 | 52.72 | 63.44 | 43.46 | 51.16 | 50.90 | 51.01   |
> | Task Arithmetic w/ DARE     | 2.84  | 68.81 | 68.38 | 54.24 | 64.38 | 42.23 | 55.15 | 49.46 | 50.68 *(-0.33)* |
> | **Task Arithmetic w/ PAVE** | 2.18  | 75.46 | 68.38 | 55.90 | 64.00 | 43.48 | 61.25 | 51.99 | **52.83 *(+1.82)*** |
> | Ties-Merging            | 5.36  | 66.28 | 68.63 | 19.09 | 63.68 | 40.44 | 55.39 | 48.74 | 45.95   |
> | Ties-Merging w/ DARE        | 6.03  | 56.42 | 68.87 | 0.61  | 66.28 | 42.11 | 54.00 | 51.26 | 43.19 *(-2.76)* |
> | **Ties-Merging w/ PAVE**    | 1.39  | 68.35 | 68.63 | 36.76 | 63.12 | 44.01 | 59.36 | 50.90 | **49.06 *(+3.11)*** |
>
> We quote the results in the discussion with other reviewer here:
>
> | ViT-B-32               | SUN397 | Cars  | RESISC45 | EuroSAT | SVHN  | GTRSB | MNIST | DTD  | Average |
> |------------------------|--------|-------|----------|---------|-------|-------|-------|------|---------|
> | Individual             | 75.3   | 77.7  | 96.1     | 99.7    | 97.5  | 98.7  | 99.7  | 79.4 | 90.5    |
> | WUDI-Merging           | 69.8  | 68.4  | 82.8    | 94.0   | 90.5 | 93.7 | 99.2 | 68.7| 83.4   |
> |  **WUDI-Merging w/ PAVE**   | 70.5  | 69.7 | 86.0    | 95.4 | 94.0 | 95.0 | 99.3 | 70.0   | **85.0 *(+1.6)***     |
> | EMR-Merging           | 75.2   | 72.8  | 93.5     | 99.5    | 96.9 | 98.1  | 99.6  | 74.4 | 88.7    |
> |  **EMR-Merging w/ PAVE**  | 76.3   | 74.3  | 93.8     | 99.3    | 96.7 | 97.6  | 99.6  | 76.1 | **89.2 *(+0.5)***     |
>
>
> Our method, PAVE, serves as a plug-and-play approach. Even though EMR-Merging is a dynamic merging method, we have also compared our approach on static and unified merging baselines, such as Task Arithmetic, Ties-Merging, and WUDI-Merging. This demonstrates that our method is general and achieves consistent improvements across both static and dynamic merging settings.

---

### Official Review · Reviewer_6qqZ · 2025-10-27

**Soundness:** 3
**Presentation:** 3
**Contribution:** 2
**Rating:** 2
**Confidence:** 4

**Summary:**

This paper introduces PAVE (Purifying and Amalgamating Task Vectors), a novel method designed to enhance model merging performance by purifying task vectors. The core premise is that task vectors, which represent the difference between a fine-tuned model and a pretrained model, often contain redundant and noisy components that degrade performance when multiple models are merged. PAVE addresses this by employing a knowledge-aware subspace analysis. It computes a task-specific covariance matrix using data from the target task and then performs Singular Value Decomposition (SVD) to decompose the task vectors. This process allows for the isolation and pruning of task-irrelevant, low-rank components, thereby retaining only the most salient, high-rank information. A key advantage of PAVE is its modular, plug-and-play design, enabling its integration with various existing model merging methods, such as EMR-Merging, Ties-Merging, and Task Arithmetic. The authors demonstrate that this purification step leads to improved performance across a range of tasks and models.

**Strengths:**

PAVE is presented as a plug-and-play method, which can benefit various existing model merging methods.
The paper is well-organized and easy to follow.

**Weaknesses:**

The strategy of retaining high-rank components is quite similar to the approach used in the TSV method [1], which also focuses on keeping the high-rank components of task vectors. It would be beneficial to compare PAVE with TSV through an ablation study, such as replacing the SVD approach with WC by directly performing SVD on the task vectors T, where T represents the task vectors.
Although the pruning strategy that selectively retains high-rank components is reasonable, the paper does not fully justify the choice of rank for each model and each layer. Further explanation of how the rank is chosen or how it varies across tasks and layers would strengthen the argument.
The theoretical grounding for the minimization objective in Equation 7 is weak. The paper fails to provide a clear justification for why minimizing this objective correlates with improved final performance. Consequently, its use as a greedy metric for selecting task parameter ranks seems arbitrary, as there is no guarantee that a larger objective value for a specific subspace corresponds to its greater importance for the task.
The performance improvements with PAVE are not consistent across methods. While it shows significant improvement with EMR-Merging and Ties-Merging, the gains with Task Arithmetic are relatively small. This variability in adaptability reduces the robustness of PAVE as a plug-and-play solution, which is a critical characteristic for practical deployment.
As shown in Figure 4, the method relies heavily on the availability of task-specific data for the decomposition process. This could be a limitation in scenarios where only limited samples or no task-specific data are available, making the method less flexible and potentially unsuitable for some use cases.
 Also, the computational cost of performing SVD on high-dimensional task vectors for large-scale models can be significant. For models with many layers and parameters, this could lead to high memory and processing requirements, which may limit the scalability of the method in practical applications.

[1] Gargiulo A A, Crisostomi D, Bucarelli M S, et al. Task singular vectors: Reducing task interference in model merging[C]//Proceedings of the Computer Vision and Pattern Recognition Conference. 2025: 18695-18705.

**Questions:**

Please see the weaknesses.

---

> ### Author Response · Authors · 2025-11-21
> **Official Comment by Authors (Part1)**
>
> Thank you for recognizing the contributions of our work and for your valuable feedback. Below, we address each of the identified weaknesses and questions in detail.
>
> **Weakness part 1:  Replace purified task vector with directly performing SVD to the task vector**
>
> Thanks for noticing us the TSV-Merging paper,  we have added it to our updated related work session. We understand the key difference between TSV-Merging and our method is that TSV-Merging directly applies plain SVD to the task vectors, whereas our method performs a knowledge-aware decomposition on the fine-tuned weights within the task vector. We provide experimental results comparing our purified task vectors with the approach of applying SVD directly to task vectors, as follows.
>
> | VIT-B-16              | SUN397 | Cars  | RESISC45 | EuroSAT | SVHN | GTRSB | MNIST | DTD  | Average |
> |-----------------------|--------|-------|----------|---------|------|-------|-------|------|---------|
> | Individual            | 79.0   | 87.0  |   96.4   |  99.1   | 97.7 |  97.7 |  99.0 | 99.7 | 92.5    |
> | EMR-Merging           | 78.5   | 82.6  | 95.4     | 99.1    | 97.6 | 98.8  | 99.6  | 78.3 | 91.2    |
> |  EMR-Merging w/ PAVE | 80.2   | 82.8  | 96.1     | 99.2    | 97.6 | 98.7  | 99.6  | 78.9 | 91.7    |
> | EMR-Merging  w/ TSV  | 78.6   | 82.7  | 95.4     | 99.2    | 97.6 | 98.8  | 99.6  | 78.3 | 91.3    |
>
>
> In conclusion, applying SVD directly to task vectors can achieve slightly better results, however, the improvement is minor and not comparable with our method's improvement.
>
>
> **Weakness part 2:  Explanation of how rank strategy assign ranks to each model and rank strategy objective**
>
>
> The rank for each model is not manually chosen nor arbitrarily searched; instead we proposed an automatic rank allocation strategy to compute the ranks. The rank strategy is designed to minimize an objective defined in Equation 9, instead of Equation 7. The design motivation of rank strategy is from the observation that fine-tuned individual models vary in their adaptation to rank pruning; some fine-tuned weights are more robust and can preserve performance even after aggressive rank pruning so require less ranks, while others are more sensitive and require more ranks. As a result, applying a naive rank strategy can lead to performance degradation.
>
> To estimate the knowledge preserving condition of pruning, we introduce the activated pruning error, defined in Equation 7. This error estimates the activated distance between our purified task vector and the original plain task vector. Since the plain task vector retains the full capabilities through fine-tuning, this error serves as a measure of how well the purified task vector preserves task-specific knowledge. To provide a more comprehensive estimation, we then normalized Equation 7 and aggregate them to form Eq. 9. Thus, minimizing the Equation 9 can improve the overall knowledge retention and improve performance.
>
> We have to correct your misunderstanding about our algorithm.
> Instead of directly searching for ranks, the greedy algorithm in our method is actually used for picking the smallest singular value in Line 9, Algorithm 1. A large objective value for either Equation 7 or Equation 9 are a sign of severe knowledge loss, hence keeping them low is important. Our rank allocation strategy is explicitly designed to minimize this objective, making it effective for allocating ranks across multiple models under constrained rank budget. For more information, please refer to Section 3.3, we have updated explanations to be clearer in this section.

---

> > ### Author Response · Authors · 2025-11-21
> > **Official Comment by Authors (Part2)**
> >
> > **Weakness part 3:  Consistency of PAVE across different methods**
> >
> > We understand PAVE shows less significant performance improvement on RoBERTa when combined with Task Arithmetic.  However, the effectiveness of the plug-in method is constrained by the strength of its underlying merging method. Task Arithmetic is the most naive approach in model merging and, unlike more advanced merging methods like emr-merging and ties-merging, task arithmetic does not have further process, like using masks to select parameters and avoid parameter overwhelming issues. Our method, PAVE, effectively reduces redundancy; however, when the base merging method itself does not address parameter overwhelming, the full benefit of PAVE cannot be revealed. As a plug-in method, we believe it is more important to see PAVE achieve significant improvement when combining with more advanced merging methods. Besides, we still achieved an encouraging +1.82% on average score in the DeBERTa experiment when combined with Task Arithmetic.
> >
> > **Weakness part 4:  Reliance on task-specific data**
> >
> > We understand your concerns. However, data access is actually allowable in the field of model merging. Some influential merging methods like Fisher-Merging [1], RegMean-Merging [2], and AdaMerging [3] also require data from the tasks to help with merging, evidence can be found in Equation 2 in [1], Equation 1 in [2] and Section 3.2.2 in [3]. And for the condition that original training data is strictly private or inaccessible, we provide two alternative solutions for it, please refer to the global comments for more details and experimental results.
> >
> > **Weakness part 5:  Memory and computational cost in application**
> >
> > The computational cost of PAVE is acceptable. All of the experiments included in our paper can be completed on a single A100 GPU. For the most computationally demanding setting in our experiments, merging Llama-2-7B models (which is a model with many layers and parameters), the highest GPU memory cost is 48G. We also report the computational time and memory use for processing a larger models, Llama-3-8B.
> >
> > | Model        | Merging Time | GPU Memory |
> > |--------------|--------------|------------|
> > | Llama-2-7B   | 32.55 min    | 48G        |
> > | Llama-3-8B   | 42.12 min       | 73G        |
> >
> > [1] Merging Models with Fisher-Weighted Averaging
> >
> > [2] Dataless Knowledge Fusion by Merging Weights of Language Models
> >
> > [3] RegMean++: Enhancing Effectiveness and Generalization of Regression Mean for Model Merging

---

> ### Author Response · Authors · 2025-11-26
> **Thanks again**
>
> Dear Reviewer 6qqZ:
>
> We would like to thank you again for your time and helpful feedback.
>
> We have tried our best to address all your comments in our response and have updated the paper accordingly. We hope you will consider raising your score based on our revisions. We would appreciate any chance to continue the discussion. Please feel free to let us know if you have any further questions or remaining concerns.
>
> Best,
>
> Authors

---

> ### Comment · Reviewer_6qqZ · 2025-11-26
>
> Thank you for the detailed and thoughtful rebuttal, as well as for addressing the earlier omission of highly relevant works such as TSV and ISO-CTS. I appreciate the inclusion of preliminary experimental comparisons with TSV-based methods. However, the current comparison seems limited to specific settings, and I believe that a more comprehensive evaluation across diverse scenarios—including varying tasks, model architectures, and data regimes—would provide a clearer picture of the relative strengths and weaknesses of your approach compared to TSV and ISO-CTS.
>
> Additionally, regarding the “knowledge‑aware” aspect of the proposed method,  this is largely achieved by leveraging access to task data. In realistic deployment scenarios, data availability is often limited or restricted, making dataless approaches—such as those employed by TSV and ISO-CTS—more broadly applicable. As the proposed approach relies on similar underlying techniques (low‑rank approximation and SVD) while additionally depending on task data, it is difficult to view the proposed pipeline as clearly advantageous or fundamentally distinct.
>
> Given these considerations, and in the absence of clear technical differentiation or substantial empirical advantages under data-restricted settings, I am inclined to maintain my original assessment.

---

> > ### Author Response · Authors · 2025-11-28
> > **Reply to Reviewer 6qqZ's Questions (Part 1)**
> >
> > Thank you for your reply. We understand that you still have some concerns, and we would like to address them with the following response.
> >
> > Q1: About comparison with TSV-Merging and ISO-CTS
> >
> > We would also like to make a more thorough comparison with these two methods.
> > However, we must notice that the numerical experiments reported in their respective papers, and also in their official implementations, are limited to merging ViT models on vision tasks. Moreover, they only report average performance without providing detailed results for each task. Please refer to Table 2 in the TSV-Merging paper and Table 1 in the ISO-CTS paper for reference.
> >
> > In contrast, our method, PAVE, has been evaluated across a broad range of merging methods, including dynamic approaches such as EMR-Merging, static approaches such as Task Arithmetic and Ties-Merging, and also recent SOTA like WUDI-Merging. PAVE has also been validated on multiple model architectures (e.g., ViT, RoBERTa, DeBERTa, and LLaMA) and across diverse tasks, including vision, natural language understanding (NLU), and natural language generation (NLG).
> >
> > In such case, the 'specific setting' (merging ViT models) mentioned by the reviewer corresponds to the only conditions used in the official implementations of those two methods. Given this, we do not believe that TSV-Merging and ISO-CTS have been adequately validated across diverse scenarios.
> >
> > As the reviewer strongly recommended comparing our method with TSV-Merging and ISO-Merging in more general settings, we conducted model merging experiments on ViT-B-32 and further adapted the TSV-Merging and ISO-Merging methods to the Llama-2-7B merging task for a fair comparison.
> >
> > | VIT-B-32               | SUN397 | Cars  | RESISC45 | EuroSAT | SVHN  | GTRSB | MNIST | DTD  | Average |
> > |------------------------|--------|-------|----------|---------|-------|-------|-------|------|---------|
> > | Individual             | 75.3   | 77.7  | 96.1     | 99.7    | 97.5  | 98.7  | 99.7  | 79.4 | 90.5    |
> > | ISO-CTS            | 71.8  | 74.2 | 88.1    | 90.5    | 82.8 | 87.7 | 97.8 | 68.3 | 82.7   |
> > | TSV-Merging            | 66.9  | 71.7 | 84.7    | 93.7   | 91.6 | 92.2 | 99.2 | 68.4 | 83.6    |
> > | WUDI-Merging           | 69.8  | 68.4  | 82.8    | 94.0   | 90.5 | 93.7 | 99.2 | 68.7| 83.4   |
> > |  WUDI-Merging w/ PAVE   | 70.5  | 69.7 | 86.0    | 95.4 | 94.0 | 95.0 | 99.3 | 70.0   | 85.0      |
> > | EMR-Merging            | 75.2   | 72.8  | 93.5     | 99.5    | 96.9  | 98.1  | 99.6  | 74.4 | 88.7    |
> > | EMR-Merging w/ PAVE   | 76.3   | 74.3  | 93.8     | 99.3    | 96.7  | 97.6  | 99.6  | 76.1 | 89.2    |
> >
> >
> > | Llama-2-7B                         | GSM8K | MATH | HuamEval | MBPP | Average |
> > |------------------------------------|-------|------|----------|------|---------|
> > | Individual                         | 54.9  | 10.7 | 39.6     | 30.1 | 33.8    |
> > | TSV-Merging                         |  23.9 | 3.4 |   22.4   | 20.7 |  17.6   |
> > | ISO-CTS                        |  21.1 | 2.6 |  17.7    | 16.5 |  14.5   |
> > | Average Merging                    | 32.2  | 5.0    | 29.2     | 29   | 23.8    |
> > | Average Merging w/ DARE            | 31.9  | 5.0    | 29.8     | 28.6 | 23.8    |
> > | Average Merging w/ PAVE   | 32.4  | 5.0    | 31.1     | 29   | 24.4    |
> >
> > As shown in the ViT merging experiments, PAVE consistently improves the performance of both WUDI-Merging and EMR-Merging. The combination of these methods with PAVE also outperforms ISO-CTS and TSV-Merging. In the Llama-2-7B merging experiment, both TSV-Merging and ISO-CTS showed weak performance on NLG tasks, while PAVE consistently enhanced the performance of simple average merging.

---

> > ### Author Response · Authors · 2025-11-28
> > **Reply to Reviewer 6qqZ's Questions (Part 2)**
> >
> > Q2: About task data depending concern
> >
> > As we have included in our global comment, when the training data is not accessible, instead, we can use data from related domain or LLM-generated synthetic data. Data from a related domain can be obtained by searching for publicly available datasets that share a similar knowledge domain with the training dataset. If such datasets are not accessible, we can also use an LLM to generate data samples in the target domain. Detailed guidance on this approach is provided in the global comment. In conclusion, in realistic deployment scenarios, data availability is not a major concern.
> >
> > Our method, PAVE, is designed to be a plug-in method, which fundamentally differs from standalone merging methods. The design of a plug-in method requires additional considerations, particularly its adaptability to a wide range of merging strategies. Therefore, directly comparing a merging method with a plug-in method is not appropriate, as they serve different roles in the model merging process. As the reviewer noted, knowledge-awareness is a key distinction between our approach and the methods the reviewer referred to. PAVE introduces a knowledge-aware decomposition framework that ensures the decomposed components are task-relevant. In contrast, the methods referenced by the reviewer rely on plain SVD, which is task-agnostic. Numerical experiments further validate the effectiveness of our method, as PAVE consistently improves the performance of both WUDI-Merging and EMR-Merging. The combined performance of these methods with PAVE also surpasses that of ISO-CTS and TSV-Merging.
> >
> >
> > Numerical Experiments:
> >
> > | VIT-B-32               | SUN397 | Cars  | RESISC45 | EuroSAT | SVHN  | GTRSB | MNIST | DTD  | Average |
> > |------------------------|--------|-------|----------|---------|-------|-------|-------|------|---------|
> > | Individual             | 75.3   | 77.7  | 96.1     | 99.7    | 97.5  | 98.7  | 99.7  | 79.4 | 90.5    |
> > | ISO-CTS            | 71.8  | 74.2 | 88.1    | 90.5    | 82.8 | 87.7 | 97.8 | 68.3 | 82.7   |
> > | TSV-Merging            | 66.9  | 71.7 | 84.7    | 93.7   | 91.6 | 92.2 | 99.2 | 68.4 | 83.6    |
> > | WUDI-Merging           | 69.8  | 68.4  | 82.8    | 94.0   | 90.5 | 93.7 | 99.2 | 68.7| 83.4   |
> > |  WUDI-Merging w/ PAVE   | 70.5  | 69.7 | 86.0    | 95.4 | 94.0 | 95.0 | 99.3 | 70.0   | 85.0      |
> > | EMR-Merging            | 75.2   | 72.8  | 93.5     | 99.5    | 96.9  | 98.1  | 99.6  | 74.4 | 88.7    |
> > | EMR-Merging w/ PAVE   | 76.3   | 74.3  | 93.8     | 99.3    | 96.7  | 97.6  | 99.6  | 76.1 | 89.2    |
> >
> > We also implemented TSV-Merging and ISO-CTS on large-scale model merging and confirmed that their performance is poor when applied to large-scale models on generative tasks. In the following numerical experiments, to address the data-restricted setting, we further included PAVE using related domain data and PAVE using synthetic data to validate our method.
> >
> >
> > | Llama-2-7B                         | GSM8K | MATH | HuamEval | MBPP | Average |
> > |------------------------------------|-------|------|----------|------|---------|
> > | Individual                         | 54.9  | 10.7 | 39.6     | 30.1 | 33.8    |
> > | TSV-Merging                         |  23.9 | 3.4 |   22.4   | 20.7 |  17.6   |
> > | ISO-CTS                        |  21.1 | 2.6 |  17.7    | 16.5 |  14.5   |
> > | Average Merging                    | 32.2  | 5.0    | 29.2     | 29   | 23.8    |
> > | Average Merging w/ DARE            | 31.9  | 5.0    | 29.8     | 28.6 | 23.8    |
> > | Average Merging w/ PAVE original   | 32.4  | 5.0    | 31.1     | 29   | 24.4    |
> > | Average Merging w/ PAVE related domain data | 32.3 | 5.1 | 31.1     | 29.4 | 24.5    |
> > | Average Merging w/ PAVE synthetic data | 32.2 | 5.1 | 30.9 | 29.3 | 24.4 |
> >
> >
> > In conclusion, we have clarified the technical differences between our method, PAVE, and TSV-Merging and ISO-CTS, and presented experimental results in broader comparison settings, demonstrating the strength of our method under data-restricted conditions.

---

### Official Review · Reviewer_3Qj9 · 2025-10-29

**Soundness:** 3
**Presentation:** 3
**Contribution:** 3
**Rating:** 6
**Confidence:** 4

**Summary:**

This paper proposes PAVE, a general plug-in method for improving model merging by removing redundancy at the task-vector level. The core steps are: for each fine-tuned model, a small number of examples are sampled from its corresponding task, layer input activations are obtained, and then used in any task-vector-based merging strategy. The paper also proposes a greedy spectral rank assignment algorithm that adaptively assigns the preserved rank to each layer/model to minimize the normalized "activation truncation error" at a given overall preservation rate. Experiments report robust performance improvements on multiple benchmarks.

**Strengths:**

1. The writing is quite easy to read and it was well-written
2.  The motivation is clear.
3.  The paper studies multiple base models
4.  The gains from their proposed method of sampling are convincing and comprehensive
5.  The method outputs a "cleaned task vector", which can directly replace the original task vector in the existing pipeline, resulting in low engineering migration costs.

**Weaknesses:**

1. The method requires sampling a certain number of training examples for each task to construct the covariance matrix. Although the authors emphasize that the number of samples can be small, this differs from strictly data-free scenarios, and the usability of the method is limited when real-world scenarios are constrained by privacy/inaccessibility of examples. The paper should discuss this more explicitly and provide degenerate behavior and alternatives for scenarios with no or very few samples.

2. The sensitivity of sample selection to task relevance was not adequately assessed: the paper points out that performance degrades when using irrelevant tasks, but does not provide quantitative guidance. In practical applications, the selection of these samples should be more clearly defined.

3. Compared to some methods based on subspace alignment or projection, PAVE focuses more on engineering and empirical evidence and lacks theoretical safeguards.

**Questions:**

1. As a plug-and-play method, this paper compares a limited number of methods. I want to know if this method can bring consistent improvements when applied to some of the latest SOTA methods, such as wudi-merging[1], tsv-merging[2], doge-merging[3], and iso-merging[4].
2. As both are plug-and-play methods, how does pave compare to awd merging[5] in performance?
3. How efficient is pave in actual computation?

[1] Whoever Started the Interference Should End It: Guiding Data-Free Model Merging via Task Vectors

[2] Task Singular Vectors: Reducing Task Interference in Model Merging

[3] Modeling Multi-Task Model Merging as Adaptive Projective Gradient Descent

[4] Task Left Behind: Isotropic Model Merging with Common and Task-Specific Subspaces

[5] Multi-task model merging via adaptive weight disentanglement

If the author can solve the question and the weakness well, i will raise my score.

---

> ### Author Response · Authors · 2025-11-21
> **Official Comment by Authors (Part1)**
>
> Thank you for recognizing the contributions of our work and for your valuable feedback. Below, we address each of the identified weaknesses and questions in detail.
>
> **Weakness 1: Concerns when training data is inaccessible**
>
> We understand your concerns. However, data access is actually allowable in the field of model merging. Some influential merging methods like Fisher-Merging [1], RegMean-Merging [2], and AdaMerging [3] also require data from the tasks to help with merging, evidence can be found in Equation 2 in [1], Equation 1 in [2] and Section 3.2.2 in [3]. And for the condition that original training data is strictly private or inaccessible, we provide two alternative solutions for it, please refer to the global comments for more details and experimental results.We understand your concerns and thank you for the comments, and we have a comprehensive explanation on this in the global comments, please refer to it.
>
> **Weakness 2: The sensitivity of sample selection to task relevance**
>
> (1) Quantitative results about sampling from irrelevant data
>
> We discussed the task relevance sample problem in Section 4.1 and illustrated the quantitative results in Figure 3. For example, in the MRPC experiments shown in Figure 3, the CO-SVD with MRPC refers to the fully knowledge-aware setting, as the covariance matrix is generated directly from the samples from the MRPC dataset. In contrast, CO-SVD with CoLA refers to an irrelevant task setting. MRPC samples are extracted from online news sources, while CoLA is drawn from books and journal articles on linguistic theory. Hence, CoLA shares limited knowledge overlap with MRPC. From the numerical experiments, we observe that under the same pruning rank, the performance of the fully task-aware setting is significantly better than that of the irrelevant task setting. The fully task-aware setting successfully activates the relevant task knowledge embedded in the LLM weights, whereas the irrelevant setting does not.
>
> (2) selection guidance
>
> In all of our numerical experiences, we do not apply specific rules for sampling and do not manually select samples. As illustrated in Figure 4, we observe that using a small number of samples (e.g., 256) is sufficient for strong performance. We also observe that the variance becomes small when the number of samples exceeds 256. This indicates that our method is robust to sampling and does not rely on specifically selected sample batches.
>
> Therefore, we can conclude quantitative results about sampling from irrelvant data in Figure 3. Results in Figure 4 further confirm that our method is robust to sampling and does not depend on manually selected examples.
>
>
>
>
>
>
> **Weakness 3: PAVE seems focus more on engineering and lack theoretical safeguards**
>
>
> Current works in this area mostly rely on empirical evidence. For example, merging methods like TSV-Merging, DOGE-Merging, Fisher-Merging [1], RegMean-Merging [2], as well as plug-in methods like DARE [4], Consensus-Merging [5] are largely evaluated through experiments. AWD-Merging seems to have some theoretical analysis; however, these analyses are limited to specific components of the method, such as demonstrating that an estimator is unbiased in probability or that a certain matrix is invertible. Comprehensive theoretical guarantees, such as a general theory validating the performance of the final merged model, remains rare.
>
> Our paper is empirically reasonable, and the effectiveness of our method is thoroughly supported by experimental results. While we considered including theoretical components prior to submission, we ultimately chose not to do so in order to maintain clarity. We believe that the lack of a complete theoretical framework should NOT be a valid reason to deny our contributions.

---

> ### Author Response · Authors · 2025-11-21
> **Official Comment by Authors (Part2)**
>
> **Question 1: Wonder if PAVE can bring consistent improvements to some latest methods**
>
>
> The applicability of our method depends on the type of merging approach. Our method is fully compatible with WUDI-Merging and yields effective improvements as shown in the results below.
>
> However, for methods like TSV-Merging, DOGE-Merging and ISO-Merging, which apply SVD directly to the full task vector, i.e. $\text{SVD}\_r(W\_{\text{FT}}-W\_{\text{B}}) = U\Sigma V^T$, our method is not compatible. These methods assume that redundant components can be eliminated through direct SVD on the task vector, often followed by additional processing steps—such as truncation and rescaling—applied to the decomposition components (e.g., $U$ and $\Sigma$). In contrast, our analysis suggests that redundant components should be activated and then removed using $\text{SVD}\_r(W\_{\text{FT}}C)C\^{-1}-W\_{\text{B}}$. Since their post-processing steps are tailored to the components from plain task vector, they are not directly applicable with our method.
>
> Notably, although these methods were released after EMR-Merging [6], they do not represent the current SOTA, which can be indicated by the following numerical results.
>
> | VIT-B-32               | SUN397 | Cars  | RESISC45 | EuroSAT | SVHN  | GTRSB | MNIST | DTD  | Average |
> |------------------------|--------|-------|----------|---------|-------|-------|-------|------|---------|
> | Individual             | 75.3   | 77.7  | 96.1     | 99.7    | 97.5  | 98.7  | 99.7  | 79.4 | 90.5    |
> | DOGE-Merging           | 70.5   | 74.8  | 88.7     | 94.1    | 91.6  | 95.7  | 99.8  | 72.5 | 85.9    |
> | ISO-Merging            | 71.8  | 74.2 | 88.1    | 90.5    | 82.8 | 87.7 | 97.8 | 68.3 | 82.7   |
> | TSV-Merging            | 66.9  | 71.7 | 84.7    | 93.7   | 91.6 | 92.2 | 99.2 | 68.4 | 83.6    |
> | WUDI-Merging           | 69.8  | 68.4  | 82.8    | 94.0   | 90.5 | 93.7 | 99.2 | 68.7| 83.4   |
> |  WUDI-Merging w/ PAVE   | 70.5  | 69.7 | 86.0    | 95.4 | 94.0 | 95.0 | 99.3 | 70.0   | 85.0      |
> | EMR-Merging            | 75.2   | 72.8  | 93.5     | 99.5    | 96.9  | 98.1  | 99.6  | 74.4 | 88.7    |
> | EMR-Merging w/ PAVE   | 76.3   | 74.3  | 93.8     | 99.3    | 96.7  | 97.6  | 99.6  | 76.1 | 89.2    |
>
>
> In conclusion, our method is compatible with WUDI-Merging and can also enhance its performance. Moreover, EMR-merging is actually still the SOTA, and our results represent a clear advancement on the SOTA performance.
>
>
>
>
>
>
> **Question 2: Comparing to AWD-Meging**
>
> We intended to include a comparison with AWD-Merging, but AWD-Merging does not offer implementation details, the official github repository shows coming soon since December 2024. Following the method description in their paper, we managed to re-implement AWD-Merging and combine it with EMR-Merging, we then compare it with the baseline EMR-Merging and EMR-Merging+PAVE.
>
> | VIT-B-32              | SUN397 | Cars  | RESISC45 | EuroSAT | SVHN | GTRSB | MNIST | DTD  | Average |
> |-----------------------|--------|-------|----------|---------|------|-------|-------|------|---------|
> | Individual            | 75.3   | 77.7  | 96.1     | 99.7    | 97.5 | 98.7  | 99.7  | 79.4 | 90.5    |
> | EMR-Merging           | 75.2   | 72.8  | 93.5     | 99.5    | 96.9 | 98.1  | 99.6  | 74.4 | 88.7    |
> | EMR-Merging w/ AWD       | 75.2   | 72.7  | 93.3     | 99.1    | 96.6 | 97.6  | 99.4  | 74.1 | 88.5    |
> |  EMR-Merging w/ PAVE  | 76.3   | 74.3  | 93.8     | 99.3    | 96.7 | 97.6  | 99.6  | 76.1 | 89.2    |
>
>
> From the table, we can see that EMR-Merging combined with PAVE outperforms both the baseline method and AWD-Merging used as a plug-in method. Additionally, our method exhibits a broader scope and higher performance, whereas AWD-Merging is limited in scope, as it relies solely on Task Arithmetic and AdaMerging [3] in their paper.
>
>
>
> **Question 3: How efficient is PAVE in actual computation?**
>
> The computational cost of PAVE is acceptable. We included a performance-computational time study in Figure 4 in our paper.  All of the experiments included in our paper can be completed on a single A100 GPU. For the more computationally demanding setting, merging LLaMA-2-7b models, the highest GPU memory cost is 48G.
>
> [1] Merging Models with Fisher-Weighted Averaging
>
> [2] Dataless Knowledge Fusion by Merging Weights of Language Models
>
> [3] AdaMerging: Adaptive Model Merging for Multi-Task Learning
>
> [4] Language Models are Super Mario: Absorbing Abilities from Homologous Models as a Free Lunch
>
> [5] Localizing Task Information for Improved Model Merging and Compression
>
> [6] EMR-Merging: Tuning-Free High-Performance Model Merging

---

> ### Comment · Reviewer_3Qj9 · 2025-11-22
> **reply to authors' rebuttal**
>
> Thanks for the authors’ reply. The response addressed some of my questions, but I still have several concerns:
>
> On data accessibility: Although some lines of work in model merging assume access to data, many widely-used merging methods are explicitly data-free. The proposed method appears to provide only marginal improvements over these data-free approaches while introducing an additional requirement for data access. Please discuss the necessity of this data access in practical scenarios.
>
> On PAVE’s actual computational cost: Requiring more than 48 GB for a 7B model seems a lot in practice. The manuscript does not present direct comparisons of computational resource between PAVE and mainstream merging methods. This makes me still confused about PAVE’s practical efficiency.
>
> On whether PAVE yields sustained improvements for recent sota: Please distinguish EMR-style merging from the static merging methods I referenced. EMR is essentially dynamic: it requires pre-merging and storing task-specific parameters and usually entails keeping substantial extra memory for those parameters at inference time. In contrast, static merging methods do not require runtime model modifications and are the more common practical setup. PAVE appears poorly compatible with most of the static approaches, which substantially limits the paper’s practical contribution.
>
> For the reasons above, I will keep the score I have currently assigned.

---

> > ### Comment · Reviewer_3Qj9 · 2025-11-22
> > **follow up on author's reply**
> >
> > I apologize for mistakenly posting my previous response under another reviewer’s section; that has now been corrected. Below is my follow-up after reading the authors’ response:
> >
> > 1. Since PAVE is positioned as a plug-and-play approach, I would expect stronger evidence of its applicability across a broader range of SOTA static merging methods. Currently, the improvement is demonstrated only on a single static SOTA method (wudi), while several other recent strong SOTA baselines are missing. This limits the generalizable contribution claimed in the paper.
> >
> > 2. Also, for convincingly demonstrating the runtime efficiency, the paper should compare the time and memory overhead of PAVE across multiple merging strategies, rather than reporting resources for a single scenario.
> >
> > 3. If PAVE is to be compared against other plug-and-play baselines (like AWD), the evaluations should likewise be conducted across multiple tasks and merging setups rather than improvements in only a single configuration.
> >
> > the current experiments might be sufficient for validating a standalone method, but the paper claims PAVE to be a plug-and-play strategy. Therefore, I think the current evaluation appears insufficient. As the authors noted, PAVE is incompatible with certain SVD-based methods (there are many works based on this); such limitations significantly constrain the contribution of the paper

---

> > > ### Author Response · Authors · 2025-11-22
> > > **Reply to Reviewer 3Qj9's Questions (Part 2)**
> > >
> > > Question 2: On computational cost
> > >
> > > In fact, most merging methods and also PAVE, can mostly run on CPU with RAM, and do not strictly require large GPU memory. In our experiments, we use GPU only for fast inferences and computation. Therefore, the implementation of PAVE can be further optimized to avoid high GPU usage.
> > >
> > > We list the computational cost with recent SOTA method WUDI-Merging to demonstrate that our computational cost is reasonable. For an 8-model ViT-B-32 merging task, WUDI-Merging requires 1.8 minutes to complete merging, while our method PAVE takes 1.7 minutes to complete and additionaly bring a +1.6 average performance improvement.
> > >
> > > | ViT-B-32                | Average Performance | Time          |
> > > |------------------------|---------------------|---------------|
> > > | WUDI-Merging           | 83.3                | 1.8 min         |
> > > | WUDI-Merging w/ PAVE   | 85.0 (+1.7)         | 3.5 min (+1.7 min) |
> > >
> > >
> > > PAVE serves as a plug-in method, meant to help baseline methods improve their results. Since the main goal of model merging is the performance of the merged model, and since merging is usually a one-time process, a plug-in method is practical if it can further improve the overall performance with a reasonable computational cost, which PAVE certainly achieves.
> > >
> > >
> > >
> > >
> > > Question 3: On whether PAVE yields sustained improvements for recent SOTA
> > >
> > > Our method is a plug-in approach that has been evaluated across a wide range of merging methods, model types, and tasks, even including challenging generative tasks. We agree that EMR-Merging is a dynamic merging method. But this again verified our method works well both for dynamic merging methods and static merging methods.
> > >
> > > Even if the reviewer decides to ignore the results on EMR-Merging, however, we have also evaluated PAVE on static merging methods such as Task Arithmetic and Ties-Merging. These results show that PAVE can still provide consistent improvements in the static setting, as we have listed in Question 1, and we also list here.
> > >
> > > | DeBERTa                     | CoLA  | SST-2 | MRPC  | STSB  | QQP   | MNLI  | QNLI  | RTE   | Average |
> > > |-----------------------------|-------|-------|-------|-------|-------|-------|-------|-------|---------|
> > > | Individual              | 59.10 | 95.06 | 89.21 | 91.31 | 91.50 | 88.52 | 93.30 | 69.31 | 84.66   |
> > > | Task Arithmeti         | 3.03  | 75.00 | 68.38 | 52.72 | 63.44 | 43.46 | 51.16 | 50.90 | 51.01   |
> > > | Task Arithmetic w/ DARE     | 2.84  | 68.81 | 68.38 | 54.24 | 64.38 | 42.23 | 55.15 | 49.46 | 50.68 *(-0.33)* |
> > > | Task Arithmetic w/ PAVE | 2.18  | 75.46 | 68.38 | 55.90 | 64.00 | 43.48 | 61.25 | 51.99 | **52.83 *(+1.82)*** |
> > > | Ties-Merging            | 5.36  | 66.28 | 68.63 | 19.09 | 63.68 | 40.44 | 55.39 | 48.74 | 45.95   |
> > > | Ties-Merging w/ DARE        | 6.03  | 56.42 | 68.87 | 0.61  | 66.28 | 42.11 | 54.00 | 51.26 | 43.19 *(-2.76)* |
> > > | **Ties-Merging w/ PAVE**    | 1.39  | 68.35 | 68.63 | 36.76 | 63.12 | 44.01 | 59.36 | 50.90 | **49.06 *(+3.11)*** |
> > >
> > > In addition, PAVE is compatible with recent strong static merging methods such as WUDI-Merging, and we show that PAVE further improves its performance. This demonstrates that our method is not limited to dynamic approaches and can be used effectively with both static and recent SOTA merging methods.
> > >
> > > | VIT-B-32               | SUN397 | Cars  | RESISC45 | EuroSAT | SVHN  | GTRSB | MNIST | DTD  | Average |
> > > |------------------------|--------|-------|----------|---------|-------|-------|-------|------|---------|
> > > | Individual             | 75.3   | 77.7  | 96.1     | 99.7    | 97.5  | 98.7  | 99.7  | 79.4 | 90.5    |
> > > | WUDI-Merging           | 69.8  | 68.4  | 82.8    | 94.0   | 90.5 | 93.7 | 99.2 | 68.7| 83.4   |
> > > |  WUDI-Merging w/ PAVE   | 70.5  | 69.7 | 86.0    | 95.4 | 94.0 | 95.0 | 99.3 | 70.0   | **85.0 *(+1.6)***     |

---

> ### Author Response · Authors · 2025-11-22
> **Reply to Reviewer 3Qj9's Questions (Part 1)**
>
> Thank you for your response. We understand that you still have some questions and concerns, and we would like to address them below. Additionally, we have updated the PDF file. At the time of your reply, the updated PDF had not yet been uploaded, so some important changes may have been missed.
>
>
>
> Question1: On data accessibility
>
> Our method is a plug-in approach that has been evaluated across a wide range of merging methods, model types, and tasks, even including challenging generative tasks. Across these diverse settings, we consistently observe meaningful improvements, so it would be inaccurate to simply conclude our gains as merely “marginal.”
>
> Below, we present numerical results to demonstrate the consistency of our improvements across various merging approaches and model types. For instance, in the first table, a merging task involving 8 DeBERTa models, we observe an average gain of +3.53 on EMR-Merging, +1.82 on Task Arithmetic, and +3.11 on Ties-Merging. Furthermore, we evaluate our method on recent SOTA WUDI-Merging; as shown in the second table, our approach yields an average improvement of +1.6 across 8 ViT model merging tasks.
>
> | DeBERTa                     | CoLA  | SST-2 | MRPC  | STSB  | QQP   | MNLI  | QNLI  | RTE   | Average |
> |-----------------------------|-------|-------|-------|-------|-------|-------|-------|-------|---------|
> | Individual              | 59.10 | 95.06 | 89.21 | 91.31 | 91.50 | 88.52 | 93.30 | 69.31 | 84.66   |
> | EMR-Merging           | 47.05 | 94.72 | 71.32 | 87.57 | 88.36 | 87.38 | 92.29 | 64.26 | 79.11   |
> | EMR-Merging w/ DARE         | 47.76 | 94.61 | 72.06 | 87.30 | 88.77 | 87.67 | 92.28 | 66.43 | 79.61 *(+0.50)* |
> | EMR-Merging w/ PAVE    | 61.32 | 94.50 | 84.31 | 89.54 | 90.34 | 87.50 | 93.01 | 60.65 | **82.64 *(+3.53)*** |
> | Task Arithmetic         | 3.03  | 75.00 | 68.38 | 52.72 | 63.44 | 43.46 | 51.16 | 50.90 | 51.01   |
> | Task Arithmetic w/ DARE     | 2.84  | 68.81 | 68.38 | 54.24 | 64.38 | 42.23 | 55.15 | 49.46 | 50.68 *(-0.33)* |
> | Task Arithmetic w/ PAVE | 2.18  | 75.46 | 68.38 | 55.90 | 64.00 | 43.48 | 61.25 | 51.99 | **52.83 *(+1.82)*** |
> | Ties-Merging            | 5.36  | 66.28 | 68.63 | 19.09 | 63.68 | 40.44 | 55.39 | 48.74 | 45.95   |
> | Ties-Merging w/ DARE        | 6.03  | 56.42 | 68.87 | 0.61  | 66.28 | 42.11 | 54.00 | 51.26 | 43.19 *(-2.76)* |
> | Ties-Merging w/ PAVE    | 1.39  | 68.35 | 68.63 | 36.76 | 63.12 | 44.01 | 59.36 | 50.90 | **49.06 *(+3.11)*** |
>
>
>
> | VIT-B-32               | SUN397 | Cars  | RESISC45 | EuroSAT | SVHN  | GTRSB | MNIST | DTD  | Average |
> |------------------------|--------|-------|----------|---------|-------|-------|-------|------|---------|
> | Individual             | 75.3   | 77.7  | 96.1     | 99.7    | 97.5  | 98.7  | 99.7  | 79.4 | 90.5    |
> | WUDI-Merging           | 69.8  | 68.4  | 82.8    | 94.0   | 90.5 | 93.7 | 99.2 | 68.7| 83.4   |
> |  WUDI-Merging w/ PAVE   | 70.5  | 69.7 | 86.0    | 95.4 | 94.0 | 95.0 | 99.3 | 70.0   | **85.0 *(+1.6)***     |
>
> The small amount of data we use is solely for computing covariance matrices that enable knowledge-aware decomposition, which our ablation studies show is crucial. When this step is replaced with a knowledge-agnostic decomposition, performance drops noticeably, demonstrating the necessity of this data-driven decomposition.
>
> It is also important to note that data usage in model merging is not unusual. For example, Fisher-Merging [1] and RegMean-Merging [2], RegMean++ [3] explicitly rely on data samples, as evidenced by Equation 2 in [1], Equation 1 in [2], and Algorithms 1 in [3],  while test-time adaptation methods like AdaMerging [4] and Concrete Subspace Learning [5] even require test data to optimizes an entropy-based objective, as shown in Section 3.2.2 in [4] and Algorithm 1 in [5].
>
> Finally, in practical, the key concern of model merging is the performance of the merged model. As long as data access is feasible, which we address in our global response and provide two alternative solutions, the performance gains offered by our method make the data requirement reasonable and worthwhile in practice.

---

### Official Review · Reviewer_nHwR · 2025-10-30

**Soundness:** 2
**Presentation:** 3
**Contribution:** 2
**Rating:** 4
**Confidence:** 4

**Summary:**

This paper presents PAVE, a plug-and-play framework for refining task vectors within a knowledge-aware subspace to enhance model merging. Departing from methods that employ random sparsification, PAVE constructs these subspaces using a context-oriented singular value decomposition (CO-SVD) guided by task-specific activations. This allows for the principled pruning of task-irrelevant components via a new spectral rank allocation strategy. PAVE's modular design enables its direct integration with existing task vector-based merging schemes. Comprehensive evaluations on GLUE benchmarks with RoBERTa and DeBERTa, alongside generative and computer vision tasks, confirm that PAVE achieves superior merging performance over strong baselines, including DARE, Task Arithmetic, and EMR-Merging.

**Strengths:**

1. The proposal to leverage task-specific activations and CO-SVD for decomposing fine-tuned weights is a principled choice. This offers a more targeted alternative to previous random or sparsity-based pruning approaches.
2. The introduction of a spectral rank allocation policy, optimizing for normalized activated pruning error across models, is a coherent and technically sound addition to enable fair pruning and avoid over-pruning critical tasks.
2. The method’s ability to serve as a plug-in for existing merging approaches (Task Arithmetic, Ties-Merging, EMR-Merging) increases its applicability and practical significance.
3. The manuscript is well-organized, the exposition is generally clear, and references to equations, figures, and tables are explicit and helpful.

**Weaknesses:**

1. The current literature review lacks the necessary depth to properly contextualize the paper's contributions. While it correctly identifies the limitations of DARE's random dropping policy, it fails to engage with a body of contemporary work exploring complementary or alternative solutions (e.g., trust-region based task vector merging, subspace boosting, and concrete subspace learning). To strengthen the paper, the authors should expand introduction to evaluate these methods, highlighting the specific challenges they do not fully address. Subsequently, this analysis should be used to more clearly frame how PAVE's knowledge-aware subspace approach offers a more effective solution. Failure to engage with them is a significant oversight, affecting the positioning of the novelty.
2. A significant practical limitation of the proposed method is its reliance on sampled activations from the training data to construct the covariance matrices. This design choice deviates from the principles of truly data-free model merging. While PAVE is not a training-based method, it is no longer strictly "data-free," a property that is highly desirable for many real-world applications where original training data may be private, proprietary, or otherwise inaccessible.
3. While PAVE demonstrates improvement on ViT models, its primary weakness is the marginal performance improvement on LLaMA, necessitating more extensive experimentation on LLMs to validate its practical utility for the generative AI landscape.

**Questions:**

1. What is the performance of PAVE when the number of tasks or models is much larger (e.g., 30)?

---

> ### Author Response · Authors · 2025-11-21
> **Official Comment by Authors (Part1)**
>
> Thank you for recognizing the contributions of our work and for your valuable feedback. Below, we address each of the identified weaknesses and questions in detail.
>
> **Weakness 1: Literature review in related fields**
>
> We have revised the related work section and added discussions covering broader related fields. Please refer to the latest PDF for details. The revised contents are marked in red in the Introduction section and also Related Work section.
>
> Concretely, we added discussion on topics including trust region based model merging, subspace boosting for model merging, and concrete subspace learning. Below, we detail the representative works corresponding to each category, outline the challenges they address, and highlight how our approach diverges from existing methods.
>
> Trust region based model merging, representative work: TATR [1].
>
> TATR is a model merging method that reduces task conflicts by constraining task vector combinations within a trust region. TATR primarily relies on task arithmetic, whereas our approach is a plug-in method that can be broadly integrated with task-vector-based merging techniques.
>
> Subspace boosting for model merging, representative work: Subspace Boosting [2]
>
> Subspace Boosting is a plug-in method that employs high order generalized singular value decomposition (HO-GSVD) to evaluate task similarity. Although HO-GSVD utilizes a shared subspace to distinguish between common and unique components across tasks, it lacks formal guarantees that the identified unique subspaces are task-aware and beneficial for merging. In contrast, our method leverages task-aware covariance to directly ensure the task-awareness of the decomposition.
>
> Concrete subspace learning, representative work: Concrete Subspace Learning [3]
>
> Concrete Subspace Learning is a test-time adaptation (TTA) method that learns a concrete mask for model merging by minimizing entropy loss. However, test-time adaptation methods are computationally intensive due to the need for additional training. In contrast, our method is training-free and inherently task-aware.
>
>
>
> **Weakness 2: Using training data to construct covariance matrix**
>
> We understand your concerns. However, data access is actually allowable in the field of model merging. Some influential merging methods like Fisher-Merging [4], RegMean-Merging [5], and AdaMerging [6] also require data from the tasks to help with merging, evidence can be found in Equation 2 in [4], Equation 1 in [5] and Section 3.2.2 in [6]. And for the condition that original training data is strictly private or inaccessible, we provide two alternative solutions for it, please refer to the global comments for more details and experimental results.

---

> ### Author Response · Authors · 2025-11-21
> **Official Comment by Authors (Part2)**
>
> **Weakness 3: Experiments on generative tasks**
>
> Model merging for generative tasks remains highly challenging. Most numerical experiments in prior merging work focus **only** on relatively small models such as RoBERTa or ViT, for example TSV-Merging [7], ISO-Merging [8], and plug-in approaches like AWD-Merging [9]. Even when generative tasks are included, as in WUDI-Merging [10], the performance is still unsatisfactory (e.g., the Table 6 in wudi-merging paper, the average performance of merging Qwen-14B only has 0.99% increase compared to the baseline). Moreover, the current plug-in method, DARE [11], is facing performance degradation in our evaluation.
>
> We quote Table 6 results in WUDI-Merging here:
>
> | Qwen-14B              | MMLU  | TruthfulQA | BBQ    | CNN   | Average |
> |-----------------------|-------|------------|--------|-------|---------|
> | Individual            | 68.35 | 53.34      | 93.53  | 19.46 | 58.67   |
> | Task Arithmetic       | 67.56 | 52.33      | 78.38  | 20.54 | 54.7    |
> | WUDI-Merging          | 69.17 | 55.71      | 80.56  | 17.33 | 55.69   |
>
>
> We quote Table 3 in our paper:
>
> | Llama-2-7B                 | GSM8K | MATH | HuamEval | MBPP | Average |
> |---------------------------|-------|------|----------|------|---------|
> | Individual                | 54.9  | 10.7 | 39.6     | 30.1 | 33.8    |
> | Task Arithmetic           | 34.3  | 4.1  | 33.5     | 27.6 | 24.8    |
> | Task Arithmetic w/ DARE   | 32.3  | 4.4  | 32.3     | 28.6 | 24.4    |
> | Task Arithmetic w/ PAVE   | 34.5  | 4.2  | 35.4     | 28   | 25.5    |
>
>
>
> In contrast, our method achieves more stable improvement than DARE and does not suffer from such degradation. Therefore, although the improvement of our method on generative tasks is not as large as those in the understanding tasks, our performance is actually an advancement of this field.
>
> **Question 1: Increase the number of tasks experiments**
>
> It is uncommon for prior works to evaluate merging 30 tasks; the only work we find claims to support 30-task ViT merging task is EMR-Merging; however, after inspecting the official github repository, the dataloader necessary for conducting the 30-task ViT merging experiment is not implemented.  Consequently, we were unable to reproduce the 30-task merging results.
>
> Most studies typically consider 14 tasks, such as TSV-Merging, ISO-Merging, and DOGE-Merging [12]. In our experiments, we adopt the ViT-B-16 merging setup from EMR-Merging, adapting it to a 14-task experiment.
>
>
> | ViT-B-16              | Vegetables | DTD   | CIFAR10 | MNIST | FOOD101 | CIFAR100 | GTRSB  | SVHN  | FashionMNIST | Pets  | EuroSAT | RESISC45 | CARS  | SUN397 | Average      |
> |-----------------------|------------|-------|---------|-------|---------|----------|--------|-------|---------------|-------|---------|----------|-------|--------|--------------|
> | Individual            | 100        | 71.7 | 89.9   | 99.2 | 87.9   | 89.8    | 95.7  | 96.2 | 93.2        | 92.2 | 99.7    | 96.4     | 87.7  | 79.0     | 91.3       |
> | EMR-Merging           | 99.9      | 58.9 | 97.0   | 98.1 | 84.8   | 89.0    | 96.1  | 90.5  | 90.2     | 86.0 | 99.0      | 95.8     | 84.4  | 80.5   | 89.3  |
> | EMR-Merging w/ PAVE  | 99.9      | 59.0 | 97.1   | 98.1 | 85.6   | 89.5    | 96.8  | 95.6 | 90.4        | 86.1 | 99.2   | 95.7    | 84.2 | 80.5  | 89.9  |
>
>
>
>
> In conclusion, our method can still achieve strong performance as the number of tasks increases.
>
> [1] Task Arithmetic in Trust Region: A Training-Free Model Merging Approach to Navigate Knowledge Conflicts
>
> [2] Subspace-boosted Model Merging
>
> [3] Concrete Subspace Learning Based Interference Elimination for Multi-task Model Fusion
>
> [4] Merging Models with Fisher-Weighted Averaging
>
> [5] Dataless Knowledge Fusion by Merging Weights of Language Models
>
> [6] AdaMerging: Adaptive Model Merging for Multi-Task Learning
>
> [7] Task Singular Vectors: Reducing Task Interference in Model Merging
>
> [8] No Task Left Behind: Isotropic Model Merging with Common and Task-Specific Subspaces
>
> [9] Multi-Task Model Merging via Adaptive Weight Disentanglement
>
> [10] Whoever Started the Interference Should End It: Guiding Data-Free Model Merging via Task Vectors
>
> [11] Language Models are Super Mario: Absorbing Abilities from Homologous Models as a Free Lunch
>
> [12] Modeling Multi-Task Model Merging as Adaptive Projective Gradient Descent

---

> ### Author Response · Authors · 2025-11-22
> **Reply to Reviewer 3Qj9 questions (part1)**
>
> Dear Reviewer nHwR,
>
> Sorry for the chaos in the comments under your review. It started with Reviewer 3Qj9, who replied to a comment under your review, which made us respond in the same thread. Now that Reviewer 3Qj9 has deleted the comment (ID: ntVvhAtj7W), we can’t delete our replies anymore.
>
> Sorry again for the chaos.

---

> ### Author Response · Authors · 2025-11-22
> **Reply to Reviewer 3Qj9 questions (part2)**
>
> This is the original reply, part 2, to the comment ntVvhAtj7W by 3Qj9. We can't delete the comment as ntVvhAtj7W has been removed.

---

> ### Author Response · Authors · 2025-11-26
> **Thanks again**
>
> Dear Reviewer nHwR:
>
> We would like to thank you again for your time and helpful feedback.
>
> We have tried our best to address all your comments in our response and have updated the paper accordingly. We hope you will consider raising your score based on our revisions. We would appreciate any chance to continue the discussion. Please feel free to let us know if you have any further questions or remaining concerns.
>
> Best,
>
> Authors

---

### Author Response · Authors · 2025-11-21
**Clarifications on Data Usage Concerns in Model Merging (Part1)**

We appreciate the reviewers’ concerns regarding potential data-privacy issues in model-merging methods, particularly the assumption that merging must be fully data-free. **We would like to clarify that data-free is not a strict requirement in the model-merging paradigm. In fact, the use of a limited number of data samples is not only widely accepted but also integral to several established merging techniques.** For example, Fisher-Merging [1] and RegMean-Merging [2], RegMean++ [3] explicitly rely on data samples, as evidenced by Equation 2 in [1], Equation 1 in [2], and Algorithms 1 in [3],  while test-time adaptation methods like AdaMerging [4] and Concrete Subspace Learning [5] even require test data to optimizes an entropy-based objective, as shown in Section 3.2.2 in [4] and Algorithm 1 in [5].


We also quote some sentences from these paper here as evidence:

Fisher-Merging, page 4:

""
In our experiments, we estimated the diagonal of the Fisher matrix via Equation 2, where $x_1, \ldots, x_N$ are drawn i.i.d. from the dataset that was used to train the model.
""

RegMean-Merging, page 16:

"In our main experiments, we use N = 1000 batches (of size 16) for computing inner product matrices. We present additional analysis about the effect of N and summarize results in Table 6. In general, performance improves as we increase N, but the performance soon saturates around N = 100."

RegMean++, page 7:

"In this section,we investigate the effects of data characteristics on merging performance with three different experiment settings: (1) number of samples: we randomly select samples from the training set of each task, (2) class imbalance: we randomly select samples from a random class in the training set of each task, and (3) OOD samples: we randomly select samples from the ImageNet database as an OOD dataset for all tasks."

AdaMerging, page 2:

"In this paper, our inspiration comes from test-time adaptation schemes aimed at optimizing model generalization when faced with previously unseen test data. Building upon these concepts, we introduce an innovative automatic unsupervised multi-task model merging scheme. This scheme leverages the minimization of prediction distribution entropy on unlabeled multi-task test data as a surrogate objective to adaptively learn model merging coefficients"


Concrete Subspace Learning, page 6:

"In this section, we discuss a situation wherein the target dataset is exclusively composed of unlabeled test data. This
implies that our objective is centered around the concept of test-time adaptation, where the focus is on adapting
and refining the merged model’s performance specifically during the testing phase."


Our method PAVE similarly relies on a small number of task-domain samples to compute the covariance matrix. Importantly, our approach does not require full access to the training set. As demonstrated in Figure 4, as few as 256 samples are sufficient to achieve strong performance and induce minor variance. To further address the reviewers’ concerns about strictly data-free settings, we additionally provide two alternative solutions that do not rely on the original training data.



**1. Using data from a related domain**

The samples we use serve only to activate the model’s weights during decomposition. If the original training data is no longer accessible, it is sufficient to use data drawn from a similar knowledge domain. Such insight has been shared in the prior study [6]. For instance, Table 7 in [6] presents an experiment where WizardLM-Evol-Instruct and Alpaca are used to collect covariance matrices, which are then evaluated on MTBench. Following this principle, we can also use data from a similar domain without access to the source training data. Concretely, in our generative task experiments under the previous setting, the covariance matrix is generated by samples from the training set of GSM8K and MATH (for math reasoning) and CodeFeedBack (for code generation). In the following experiments, we show that Lila-Math (for math reasoning) and OpenCodeInstruct (for code generation) can effectively replace samples from the original training sets. We include numerical results below to demonstrate this.

---

> ### Author Response · Authors · 2025-11-21
> **Clarifications on Data Usage Concerns in Model Merging (Part2)**
>
> **2. Using LLM-generated synthetic data**
>
> When even related-domain datasets are unavailable, one can use synthetic data generated by large language models in the same knowledge domain. In practice, the prompt for synthetic data generation can be composed of a basic task description + a knowledge domain description + a few examples. The basic task description helps the language model recognize that the prompt concerns a sample generation task, while the knowledge domain description provides information of the knowledge domain, and a few examples can improve the consistency of generation. Such sythetic data is sufficient to activate the relevant parameter subspaces for our decomposition procedure.
>
>
> Numerical Experiments:
>
> | Llama-2-7B                         | GSM8K | MATH | HuamEval | MBPP | Average |
> |------------------------------------|-------|------|----------|------|---------|
> | Individual                         | 54.9  | 10.7 | 39.6     | 30.1 | 33.8    |
> | Average Merging                    | 32.2  | 5.0    | 29.2     | 29   | 23.8    |
> | Average Merging w/ DARE            | 31.9  | 5.0    | 29.8     | 28.6 | 23.8    |
> | Average Merging w/ PAVE original   | 32.4  | 5.0    | 31.1     | 29   | 24.4    |
> | Average Merging w/ PAVE related domain data | 32.3 | 5.1 | 31.1     | 29.4 | 24.5    |
> | Average Merging w/ PAVE synthetic data | 32.2 | 5.1 | 30.9 | 29.3 | 24.4 |
>
>
>
> Therefore, data access is actually not forbidden in the area of model merging [1,2,3,4]. When the source training data is strictly inaccessible, one can easily use dataset in similar knowledge domain or use llms to generate synthetic data in similar knowledge domain, and still achieve good performance using our method.
>
>
> Reference:
>
> [1] Merging Models with Fisher-Weighted Averaging
>
> [2] Dataless Knowledge Fusion by Merging Weights of Language Models
>
> [3] RegMean++: Enhancing Effectiveness and Generalization of Regression Mean for Model Merging
>
> [4] AdaMerging: Adaptive Model Merging for Multi-Task Learning
>
> [5] Concrete Subspace Learning Based Interference Elimination for Multi-task Model Fusion
>
> [6] CorDA: Context-Oriented Decomposition Adaptation of Large Language Models for Task-Aware Parameter-Efficient Fine-tuning

---

### Meta-Review · Area_Chair_4Y24 · 2026-01-01

**Summary:**

This paper proposes PAVE, a plug-and-play approach for purifying task vectors in a knowledge-aware subspace to improve model merging. The core idea of leveraging task-specific activations and context-oriented SVD to remove redundant components is intuitive and reasonably motivated, and several reviewers agree that the method is technically sound and clearly presented. The empirical evaluation is fairly extensive, covering multiple backbones, tasks, and merging strategies, and the authors demonstrate consistent performance improvements in a number of settings, particularly when combined with EMR-Merging and other task-vector-based approaches. The rebuttal is thorough and addresses most reviewer questions with additional experiments and clarifications.

However, despite these strengths, there remain substantive concerns regarding the scope, positioning, and practical impact of the contribution. A central point of contention is the reliance on task-domain data to construct covariance matrices. While the authors correctly note that some prior merging methods also assume limited data access, this requirement nonetheless weakens the appeal of the method in scenarios where data-free merging is a key motivation. The paper does not convincingly justify why the added data dependency is necessary relative to the relatively modest gains observed in some settings, particularly for large-scale generative models. In addition, although PAVE is framed as a plug-and-play method, its incompatibility with several recent and widely discussed static SVD-based merging approaches significantly limits the breadth of its applicability. As a result, the claimed generality of the method is somewhat overstated.

There are also concerns about scalability and practicality. The SVD-based decomposition over high-dimensional weights introduces non-trivial computational and memory overhead, and while the authors argue that merging is typically an offline process, the cost may still be prohibitive for larger models and broader adoption. Finally, while the empirical results are comprehensive, the paper relies almost entirely on experimental validation and offers limited theoretical insight into why the proposed rank allocation objective reliably correlates with downstream merging performance, leaving some aspects of the method insufficiently grounded.

Overall, the paper presents a reasonable and incremental improvement over existing task-vector-based merging techniques, but the limitations in applicability, data assumptions, and scalability prevent it from constituting a sufficiently strong or broadly impactful contribution for acceptance at ICLR.

**Reviewer Concerns:**

The rebuttal clarified several points, including the method design, related work positioning, and additional experimental results, and partially addressed concerns about data usage and robustness to sample selection. However, key concerns remain. The reliance on task-domain data continues to limit the method’s appeal in data-free merging scenarios, and the rebuttal does not fully justify this requirement given the modest gains in some settings. In addition, the incompatibility with several recent static SVD-based merging methods weakens the plug-and-play claim, and concerns about scalability and limited theoretical grounding remain insufficiently resolved.

**Reviewer Scores:**

Reviewer nHwR did not participate in the discussion phase. Based on the tone of the initial review and the nature of the rebuttal, it is likely that some clarifications (e.g., related work coverage and additional experiments) could have slightly improved their perception, but the core concerns regarding data usage and scalability would likely remain. I therefore do not expect a significant score change.

Reviewer 3Qj9 actively engaged during the discussion and explicitly stated that they would keep their assigned score after reading the rebuttal. As such, no score change is expected.

Reviewer 6qqZ maintained strong reservations throughout the review regarding data dependency, scalability, and methodological justification. While the rebuttal addressed these points in detail, it is unlikely that their overall assessment would change materially. I do not expect an increase in score.

---

### Decision · Program_Chairs · 2026-01-26

Reject